# Transfer of motor learning is associated with patterns of activity in the default mode network

Ali Rezaei[1], Corson N. Areshenkoff[1,2], Daniel J. Gale[1], Anouk J. De Brouwer[1], Joseph Y. Nashed[1], J. Randall Flanagan[1,2], Jason P. Gallivan[1,2,3]*

1 Center for Neuroscience Studies, Queen's University, Kingston, Canada, 2 Department of Psychology, Queen's University, Kingston, Canada, 3 Department of Biomedical and Molecular Sciences, Queen's University, Kingston, Canada

* gallivan@queensu.ca

## Abstract

An often-desired feature of motor learning is that it generalizes to untrained scenarios. Yet, how this is supported by brain activity remains poorly understood. Here we show, using human functional MRI and a sensorimotor adaptation task involving the transfer of learning from the trained to untrained hand, that the transfer phase of adaptation re-instantiates a highly similar large-scale pattern of brain activity to that observed during initial adaptation. Notably, we find that these neural changes, rather than occurring at the level of sensorimotor regions, predominantly occur across distributed areas of higher-order transmodal cortex, specifically in regions of the default mode network (DMN). Moreover, we show that these learning-related neural changes relate to the structural properties of transmodal cortex (its myelin content and neurotransmitter receptor density), and that intersubject differences in DMN activity relate to both adaptation- and transfer-phase task performance. Together, these findings suggest that the transfer of learning across the hands is supported by the re-expression of the same activity patterns in the DMN as those that support initial learning. Collectively, these results offer a unique characterization of the whole-brain macroscale changes associated with sensorimotor learning and generalization, and establish a key role for higher-order brain areas, such as the DMN, in the transfer of learning to untrained scenarios.

## Introduction

The central nervous system's ability to adapt our movements to new and changing environments is fundamental to daily life. This adaptability not only involves mastering movements within specific trained contexts but also extends to transferring learned skills to novel scenarios, a phenomenon known as generalization [1,2]. Understanding how our motor memories generalize to new situations, such as using one's non-dominant hand for a well-practiced task, provides crucial insights into the

**Data availability statement:** All behavioral and imaging data (including T1w and functional scans) are freely available at the following repository at OpenNeuro (https://openneuro.org/datasets/ds005598).

**Funding:** This work was supported by operating grants from the Canadian Institutes of Health Research Grant (https://cihr-irsc.gc.ca/e/193.html; grant number: PJT175012) and the Natural Sciences and Engineering Research Council (https://www.nserc-crsng.gc.ca/index_eng.asp; grant number: RGPIN-2017-04684), awarded to JPG. The funders had no role in study design, data collection and analysis, decision to publish, or preparation of the manuscript.

**Competing interests:** I have read the journal's policy and the authors of this manuscript have the following competing interests: JPG and DJG are employees of Voxel AI Inc. The other authors report no conflicts of interest.

**Abbreviations:** AC-PC, anterior and posterior commissure; BOLD, blood oxygenation level-dependent; CSF, cerebrospinal fluid; DCM, Dynamic Causal Modeling; DMN, default mode network; FC, functional connectivity; FD, framewise displacement; FDR, false discovery rate; GLM, General Linear Model; GM, gray matter; IFG, inferior frontal gyrus; INU, intensity non-uniformity; mPFC, medial prefrontal cortex; PC, principal component; PCA, principal component analysis; PCC, posterior cingulate cortex; PC1, First Principal Component; rmANOVA, repeated measures ANOVA; RSA, Representational Similarity Analysis; RSM, representational similarity matrix; SEM, standard error of the mean; UMAP, uniform manifold approximation; VMR, visuomotor rotation; WM, white matter.

neural bases that support behavior and has significant implications for patient treatment and rehabilitation [3–6]. Yet, the specific neural processes that underpin motor generalization remain unresolved.

Understanding these processes requires examining not just isolated brain regions, but how large-scale neural systems dynamically coordinate their activity. Traditional fMRI approaches often focus on localized activation changes or connectivity between specific pairs of regions. However, motor learning and generalization engage widely distributed circuits spanning cortical, subcortical, and cerebellar areas [1,7–10]. Investigating changes within such distributed systems benefits from methods that capture the holistic organization of brain-wide functional organization. To achieve this, we employed manifold learning, a powerful dimensionality reduction technique [11–13]. This approach, which builds upon key observations from both neuronal [11, 14–16] and fMRI [17–21] studies, embeds the unique functional connectivity (FC) profile of each brain region (its pattern of covariance with all other regions) into a common low-dimensional space. The resulting 'manifold' provides a representation of the brain's overarching functional geometry, revealing the principal axes along which functional network organization varies [13,21]. By analyzing how regions are positioned within this manifold across different task phases, we can track large-scale network reconfigurations—identifying shifts in functional integration and segregation—that underpin learning and its transfer [22–26].

One well-studied form of motor generalization in humans examines how visuomotor adaptation of reaching movements transfers across the hands. Such adaptation is known to involve at least two parallel processes: An automatic, implicit process that adapts gradually and a strategic, explicit process that adapts rapidly [27–30]. While the implicit process is thought to involve the recalibration of internal models within sensorimotor cortex and the cerebellum, the explicit process is associated with the conscious application of strategies and is hypothesized to engage higher-order cognitive regions [29–32]. Critically, behavioral studies have indicated that the successful transfer of visuomotor adaptation from the trained to the untrained hand is predominantly mediated by explicit strategy use rather than implicit recalibration [28,33–36]. This is noteworthy as recent computational work indicates that context—the specific set of environmental and internal cues associated with a given experience, which is often highly explicit in nature—plays a pivotal role in how the brain manages and generalizes learning [37–39].

Despite the recognized importance of explicit strategies in motor learning and generalization, the specific neural substrates supporting these processes are not fully understood. The default mode network (DMN)—a collection of interconnected brain areas within higher-order transmodal cortex—has been implicated in various explicit cognitive functions, including memory retrieval, planning, and decision-making [40–42]. While traditionally associated with internally focused tasks such as autobiographical memory and future thinking [40,43–45], emerging evidence suggests that the DMN may also play a role in tasks requiring the formation of strategies [46–48]. The DMN's unique neuroanatomical and neurophysiological characteristics may enable it to support flexible strategy use: it occupies the top position within the brain's

cortical processing hierarchy, allowing it to integrate information across diverse cognitive and sensory systems [13,48,17]; its regions exhibit prolonged temporal integration windows, which may facilitate the maintenance and implementation of strategies over extended periods [17,49–51]; and its high receptor density and complex dendritic morphology suggest a heightened capacity for computational flexibility and learning [17,52,53]. Given that explicit cognitive strategies are crucial for the transfer of visuomotor adaptation, it is plausible that the DMN uniquely contributes to this process, perhaps reflected by specific changes in its manifold embedding.

To explore this idea, we investigated changes in the manifold structure derived from FC patterns across cortical, sub-cortical, and cerebellar brain regions during sensorimotor adaptation and generalization. In our study, human participants learned to adapt their reaching movements during a visuomotor adaptation task performed with their right hand (learning phase), prior to performing the same task with their untrained left hand (transfer phase). To characterize learning- and transfer-related changes at the whole-brain level, using our manifold embedding approach, we examined the extent to which different brain regions shifted their connectivity profiles across different phases of sensorimotor adaptation and generalization.

By disentangling effector-specific versus effector-independent changes in manifold structure, we found that the con-nectivity of DMN areas, rather than sensorimotor regions, was predominantly modulated across both learning and transfer phases. Specifically, we found that the transfer phase of learning re-expressed a highly similar pattern of changes in DMN manifold structure as were observed during initial adaptation. These results suggest that the transfer of learning across the limbs is supported by the re-instantiation of the same activity patterns that support initial learning. Together, these findings offer new insights into the role of higher-order brain networks, such as the DMN, in real-world motor behavior, and contribute to a more comprehensive understanding of the neural substrates underlying the transfer of motor learning.

## Results

To study the neural processes that support sensorimotor adaptation and generalization, we had human participants (*N*=38) perform a classic visuomotor rotation (VMR) task [54] in the MRI scanner. In this task, participants were required to move a cursor, representing their finger position, to hit a target that could appear in one of eight locations (along an invisible ring), using an MRI-compatible touchpad (Fig 1A). Participants began the study by performing two separate 'baseline' blocks (64 trials each; 6 s per trial)—the first with their non-dominant left hand (LH Baseline) and the second with their dominant right hand (RH baseline)—in which the motion of the cursor directly matched their finger movements. Following these baseline blocks, subjects performed a 'learning' block with their right hand (160 trials; RH Learning) in which the relationship between finger movement on the touchpad and movement of the cursor was rotated clockwise by 45°. Consequently, subjects had to learn to adapt their hand movements in a 45° counterclockwise direction to accurately hit the target. Following this RH learning block, participants performed 16 "report" trials, which allowed us to obtain a direct index of subjects' explicit strategy in counteracting the VMR [28]. In these report trials, subjects were asked to indicate their aim direction using a dial on a joystick, controlled by their left hand, prior to executing a target-directed movement via their right hand. Finally, following these report trials, subjects completed the testing session by performing the exact same learning block (80 trials) but using their left hand (LH Transfer), allowing us to examine the intermanual transfer (i.e., generalization) of sensorimotor adaptation to the untrained hand.

As shown in Fig 1B, we found that subjects, on average, were able to reduce their visuomotor errors during each of the learning and transfer periods. Moreover, consistent with prior work on the generalization of sensorimotor adaptation across the hands [28,34,55], we found that subjects on average exhibited lower initial errors during the transfer than learn-ing phase (see right inset in Fig 1B). This result has been taken to indicate that the neural representation of learning is to some extent shared across the two hands (i.e., is effector-independent [27,56,57]).

Next, in order to explore the changes in brain activity that underpin sensorimotor adaptation and generalization, we examined task-related FC within six distinct, equal-length 32-trial epochs (i.e., 96 imaging volume windows) over the time

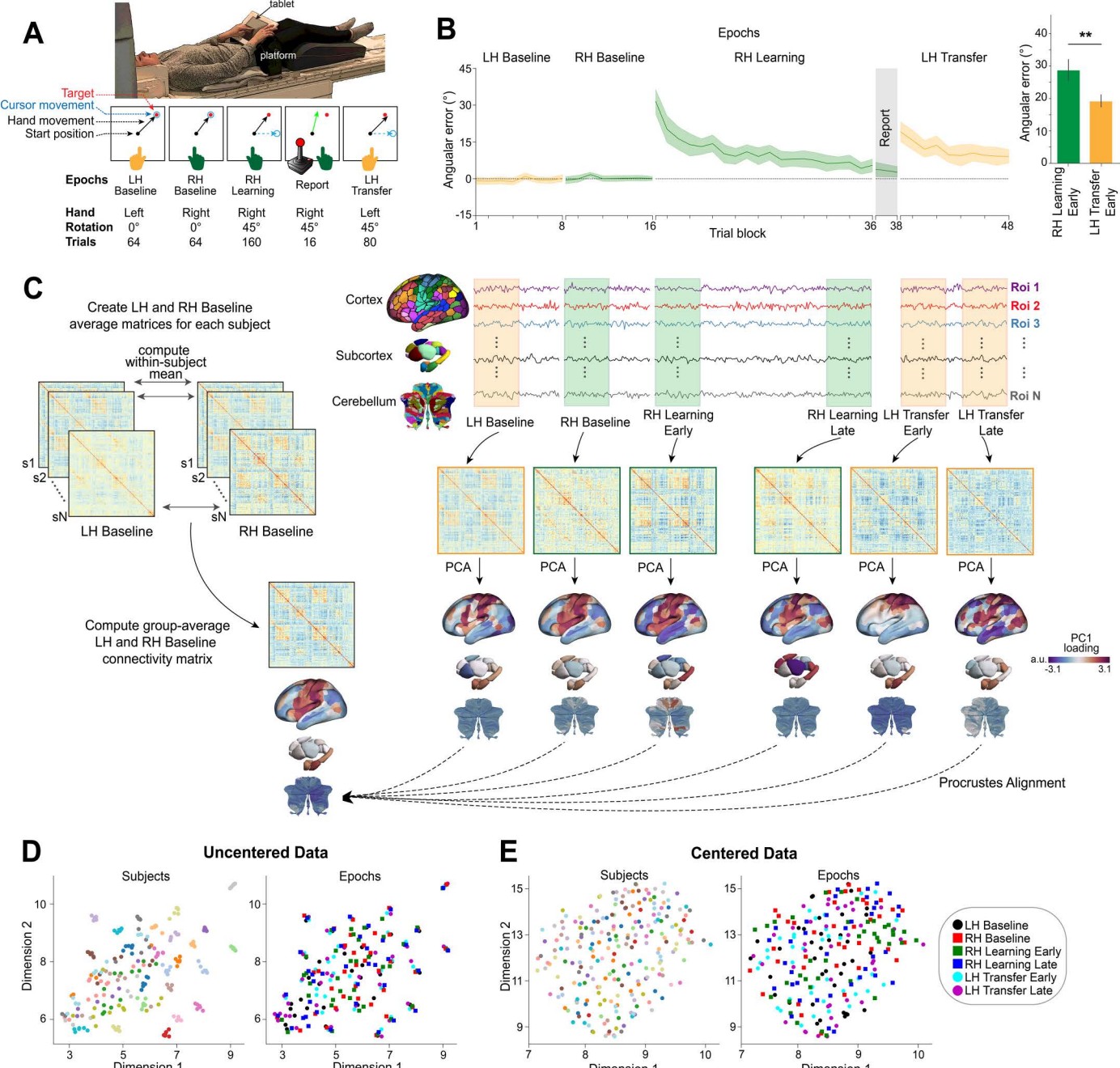

**Fig 1. Task structure and overview of fMRI analysis. (A)** MRI setup (top) and task structure (bottom). **(B)** Average participant learning curves (median across 8-trial bins). Orange and green traces denote periods during the task in which the left hand (LH) and right hand (RH) were used, respectively. Banding around each trace denotes ±1 standard error of the mean (SEM). Inset bar graphs at right compare the initial error (first 16 trials) for RH learning and LH transfer trials. ** denotes $p < 0.01$ **(C)** Neural analysis approach. *Right*, for each participant, we extracted time series data from the Schaefer 400 cortical parcellation, Tian 32-region subcortical parcellation, and Nettekoven 32-region cerebellar parcellation for six equal-length task epochs (indicated by the coloured boxes) and then computed functional connectivity (FC) matrices for each epoch. We then estimated connectivity manifolds for each task epoch using PCA with centered and thresholded connectivity matrices (see Materials and methods). To visualize how the dominant pattern of FC changed across task phases before alignment to a common space, the surface brain maps depict the first Principal Component (PC1) loadings calculated from a representative subject's covariance matrices for each specific task epoch (e.g., LH Baseline, RH Baseline, etc.). The color bar indicates the PC1 loading value for each brain region, reflecting the strength and direction of that region's contribution to the epoch's primary connectivity pattern. These maps serve an illustrative purpose and will differ slightly from the first PC of the common template Baseline manifold shown in [Fig 2](link),

which provides the reference space for all subsequent alignment and analyses. *Left*, construction of this template Baseline manifold. All manifolds were aligned using Procrustes transform to a common template manifold created from a group-averaged FC matrix based on the mean across the LH and RH Baseline epochs. This allowed us to assess learning-related changes in manifold structure from this Baseline architecture. **(D and E)** Subject-level clustering is abolished through a Riemannian centering approach. UMAP visualization of the similarity of FC matrices, both before centering (*D*) and after (*E*) centering. In these plots, each point represents a single FC matrix, color-coded either to subject identity (left panels) or task epoch (right panels), with its location in the two-dimensional space based on the similarity between matrices. Note that the uncentered connectivity matrices in D show a high degree of subject-level clustering, thus obscuring any differences in task structure. By contrast, the Riemannian manifold centering approach (in E) abolishes this subject-level clustering. The data and code needed to generate this figure can be found in https://zenodo.org/records/15648991.

course of the task. We defined the LH and RH Baseline epochs as the first four 8-trial bins (32 trials) of each functional scan, respectively, and defined 'Early' and 'Late' learning epochs as the first and last 32 trials of each of the RH Learning and LH Transfer functional scans, respectively. For each subject and each of these six epochs, we extracted region-wise mean blood oxygenation level-dependent (BOLD) time series data from the Schaefer 400 cortical parcellation [58], Tian 32-region subcortical atlas [59], and Nettekoven 32-region cerebellar atlas [60]. From this data, we then estimated covariance (FC) matrices for each epoch (LH Baseline, RH Baseline, RH Learning Early, RH Learning Late, LH Transfer Early, and LH Transfer Late; see bottom Fig 1C; [for a similar approach, see [22,24]).

Prior work [61,62], including our own [22,24,32,63], has shown that the majority of variance in patterns of whole-brain FC can be attributed to static, subject-level differences, and that these trait-like, subject-level features can mask any task-related (state-dependent) differences in brain activity. To mitigate this issue, using established procedures, we centered all subjects' connectivity matrices using the Riemannian manifold approach [22,32,63,64] (see Materials and methods). To demonstrate the impact of this centering method and its importance for being able to extract any learning-related effects within our data, we projected all subjects' FC matrices, both before and after our centering procedure, using the unsupervised nonlinear dimensionality-reduction technique uniform manifold approximation (UMAP; [65]). In this UMAP space, nearby points represent more similar patterns of whole-brain covariance compared to distant points. As shown in Fig 1D, prior to the centering procedure, nearly all the FC matrices cluster according to subject identity. This observation aligns with previous work using the Midnight Scan Club dataset [61] showing that subject-level structure can impede the detection of any task-related effects. However, after use of our centering procedure, this subject-level clustering is completely eliminated (Fig 1E), thus allowing us to explore modulations in connectivity patterns associated with the different task epochs.

Next, to investigate changes in whole-brain connectivity during the learning task, we used the centered matrices to estimate whole-brain (cortical, subcortical, and cerebellar) connectivity manifolds for each participants' FC matrices from the six epochs [see also 22,24]. Following from prior work [12,66,67], each matrix was initially transformed into an affinity matrix by calculating the pairwise cosine similarity between regions after row-wise thresholding (see Materials and methods). Next, we applied dimensional reduction to the affinity matrices using principal component analysis (PCA) to obtain a set of principal components (PCs) that provide a low-dimensional representation (i.e., manifold) of cortical-subcortical-cerebellar connectivity structure for each epoch. The right panel in Fig 1C shows the spatial pattern of the first PC (PC1) loadings derived from the centered FC matrix of a representative subject for each task epoch. Each brain region's color indicates the loading value (see Fig 1C caption), illustrating shifts in the primary axis of connectivity organization across the task phases. Next, to allow for direct comparison across epochs, we used Procrustes transformation to align each manifold onto a template Baseline manifold, constructed from the mean of all LH Baseline and RH Baseline connectivity matrices by first averaging within, and then across, all subjects (Fig 1C, left). This approach served two purposes: First, the common Baseline manifold served as a reference for manifold alignment [12], enabling direct comparisons among all participants within a single, unified task-based neural space. Second, it facilitated the identification of any learning- and transfer-related deviations from this baseline manifold architecture [see also 22,24].

## Whole-brain manifold structure during Baseline trials

The top three PCs that comprise the template Baseline manifold (Fig 2A) describe the overarching functional organization of whole-brain activity during Baseline trials with the left and right hand. PC1 mainly distinguishes cortical sensorimotor regions (positive loadings in red) from visual cortex and the cerebellum (negative loadings in blue; see also Fig 2C). Meanwhile, PC2 mainly separates visual, somatomotor and cerebellar regions (in red) from higher-order transmodal, DMN regions (in blue; see also Fig 2C). Finally, PC3 primarily distinguishes visual and posterior cingulate regions (in red) from the cerebellum and subcortex (in blue; see also Fig 2C). Collectively, these top three PCs explain the majority of the total variance (i.e., 50.90%) in the Baseline FC data (Fig 2B), and we retained these top three PCs for our further analyses. [However, note that the inclusion of PCs 4 and 5 in our analysis, which explained 9.68% and 5.87% of the variance, respectively, did not significantly impact any of our findings; see Fig A in S1 Text].

We then aimed to characterize the embedding of each cortical, subcortical, and cerebellar brain region in this Baseline-derived manifold space, such that we could then assess subsequent shifts in the positioning of these regions during the learning and transfer phases of the task. To this end, following from previous methods [22–24], we computed the Euclidean distance of each region from the manifold origin (coordinates [0,0,0]; see Fig 2D). This measure, which we term 'Manifold Eccentricity', provides us with a multivariate index of each brain region's relative positioning in the three-dimensional manifold space (Fig 2E). From this whole-brain manifold perspective, regions with higher eccentricity (e.g., located at the extremes of the manifold) can be interpreted as generally having greater functional segregation from other networks in the brain, whereas regions with lower eccentricity (e.g., situated near the manifold's center) can be interpreted as generally having higher functional integration (lower segregation) with other networks [23,68,69]. Congruent with this interpretation, we found that our eccentricity measure was significantly correlated with several classic graph theoretical measures of integration and segregation (Fig B in S1 Text). Thus, changes in manifold eccentricity during sensorimotor adaptation and generalization provide insight into the functional segregation versus integration of different brain regions over time.

## Global changes in manifold structure during learning and transfer

We next sought to establish changes in manifold architecture across the different phases of the task. To this end, we first applied multivariate analytical methods to investigate, at the global whole-brain level, potential differences in manifold eccentricity across epochs. Fig 3A visualizes the mean (across subject) whole-brain eccentricity for each of the six task epochs using two-dimensional UMAP [71]. In this two-dimensional projection, nearby points represent epochs with more similar whole-brain eccentricity patterns. This visualization suggests that multiple types of task-related information might be encoded in the global manifold structure. For instance, UMAP dimension 1 suggests that relevant task epoch-related information is being encoded in the whole-brain eccentricity data, with LH and RH Baseline trials (black dot and red square) represented differently from the LH Transfer epochs (cyan and purple dots) and the RH Learning epochs (green and blue squares; read Dimension 1 from left to right). By contrast, UMAP dimension 2 also suggests that, independent of the hand being used, the early learning and early transfer epochs of the task (cyan and green dots) are represented differently from both the Baseline and Late Learning epochs (read Dimension 2 from top to bottom). Together, this suggests a multiplexing of both effector and effector-independent information in the manifold eccentricity data.

To complement these UMAP visualizations and quantitatively test specific hypotheses about the information represented, we performed Representational Similarity Analysis (RSA; [72,73]) on the eccentricity data. Using established methods [74] we constructed, for each subject and epoch, a representational similarity matrix (RSM) by calculating the Pearson correlation between the vectors of regional eccentricity values (464 regions) for every pair of the six task epochs (Fig 3B shows the group-averaged RSM). This RSM captures the pairwise similarity between the neural states associated with each epoch based on their whole-brain eccentricity patterns.

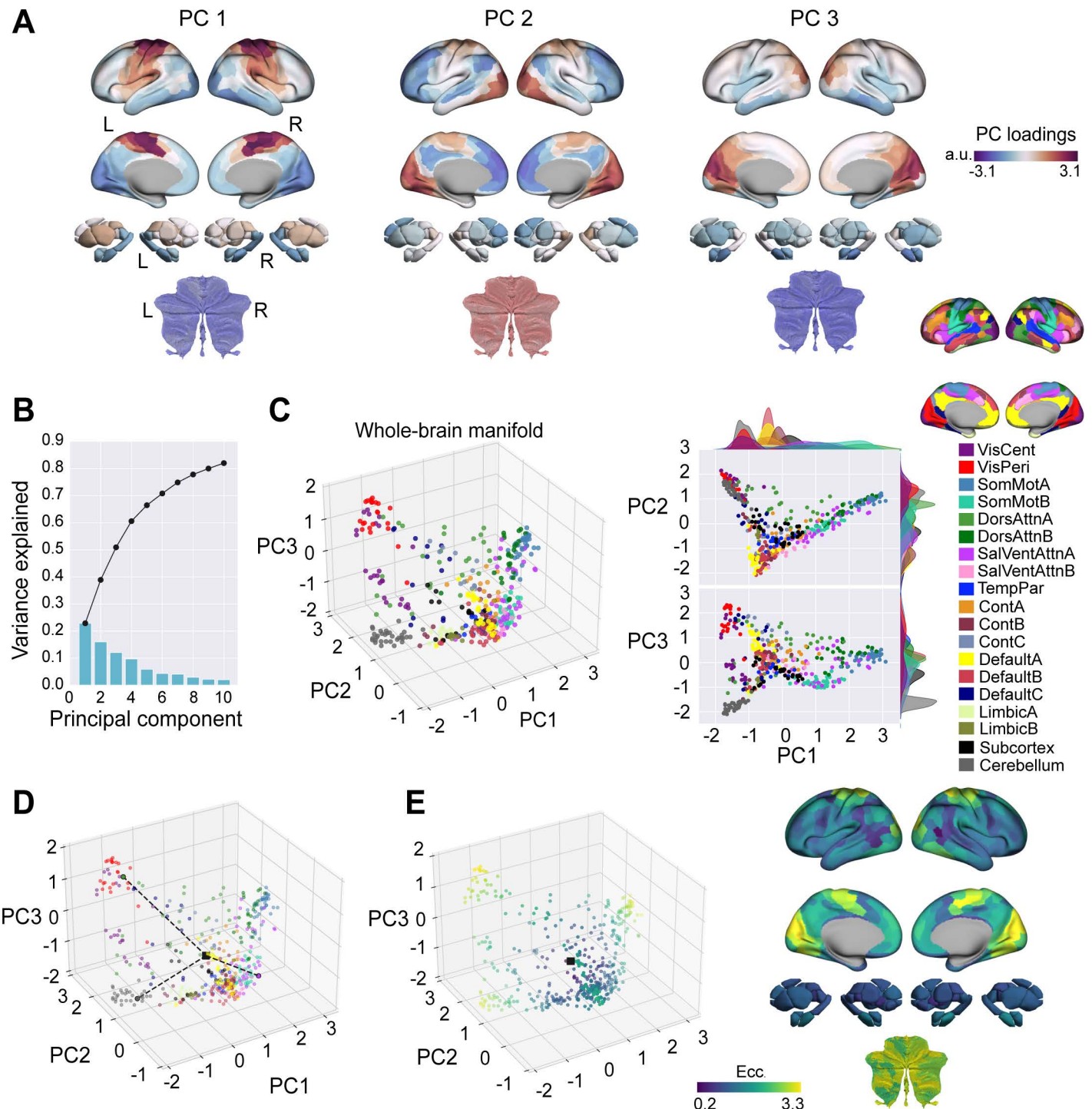

**Fig 2. Main connectivity axes extracted from Baseline trials. (A)** Region loadings across cortex (top), subcortex (middle) and cerebellum (bottom) for the top three PCs. **(B)** Variance explained for the first 10 PCs. Black trace shows the cumulative variance explained across PCs, whereas the blue bars denote the variance explained for each individual PC. **(C)** The Baseline (template) manifold in low-dimensional space, with regions colored according to the Yeo and colleagues, 17-network assignment [58,70]. **(D)** Illustration of how eccentricity is computed. A single region's eccentricity along the manifold is calculated as the Euclidean distance (dashed line) from manifold centroid (black square). The eccentricity of three example brain regions is highlighted. **(E)** Regional eccentricity during Baseline trials. Each brain region's eccentricity is color-coded in the low-dimensional manifold space (*left*) and on the cortical, subcortical, and cerebellar surfaces (*right*). Black square denotes the center of the manifold (manifold centroid). L = left; R = right.

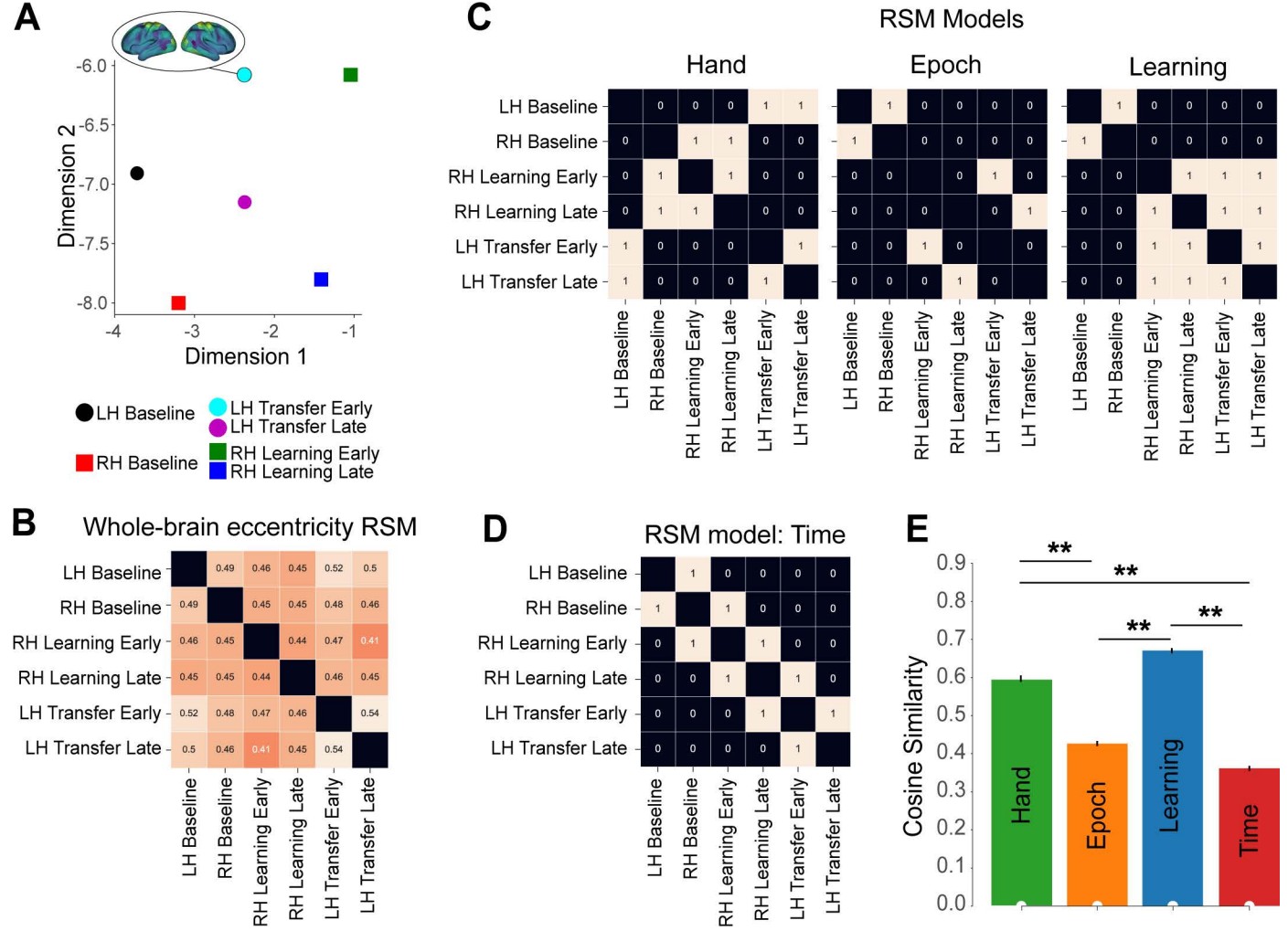

**Fig 3. Multiplexing of task-based information in global manifold architecture. (A)** UMAP visualization of the similarity in neural states across task epochs (based on the data in B). In this plot, each point represents the across-subject mean whole-brain eccentricity for different task epochs (see legend at bottom), with nearby points representing more similar patterns of whole-brain eccentricity. Note that the two UMAP dimensions appear to capture different types of information about the task structure (see text). **(B)** Average across-subject representational similarity matrix (RSM) for the patterns of whole-brain eccentricity associated with the six task epochs (the value in each cell represents the Pearson spatial correlation). Note that the RSM is symmetric about the diagonal (black squares, which represent self-correlations). **(C)** Idealized model RSMs representing different hypotheses about task-related structure. **(D)** An idealized model RSM representing a control hypothesis based on the passage of time. In these model RSMs (C and D), values of 1 indicate predicted similarity between epochs according to the model, while 0 indicates predicted dissimilarity. These models are compared to the empirical data RSM shown in B. **(E)** Model comparisons. Each bar indicates the across-subject mean of cosine similarity between the RSM of the whole-brain eccentricity data (in B) and each of the models (in C and D; higher values mean higher similarity). Inference was performed via bootstrap resampling of subjects (1,000 bootstrap samples). Error bars indicate the standard error of the mean. The significance of each model for a one-sided comparison against zero is marked by white dew drops on the horizontal axis, and against the lower-bound estimate of the noise ceiling (gray icicles; FDR-corrected for 4 models). Models' Pairwise differences are summarized by stars. ** denotes $q < 0.05$. The data and code needed to generate this figure can be found in https://zenodo.org/records/15648991.

We next tested how well this empirical RSM structure (Fig 3B) was explained by three different *a priori* theoretical models representing distinct hypotheses about task structure (Fig 3C and 3D): The first model predicted the encoding of hand information only ('Hand' model, Fig 3C, left), the second predicted the encoding of the task epoch only (Baseline, Early or Late; 'Epoch' model, Fig 3C, middle), and the third predicted the encoding of the Baseline versus Learning periods only

('Learning model'; Fig 3C, right). These binary model RSMs represent theoretical similarity structures (where 1 denotes predicted similarity, 0 predicted dissimilarity) corresponding to each hypothesis. We compared each participant's empirical RSM to these theoretical model RSMs using cosine similarity, with higher values indicating a better fit between the model and the eccentricity data. The group-averaged results are shown in Fig 3E. Statistical inference via bootstrapping revealed that both the 'Hand' and the 'Learning' model performed better than the 'Epoch' model, with the 'Learning' model performing best (though not statistically better than the Hand model; see Fig 3E).

Note that one potential explanation of these above results is that they could simply reflect the passage of time; i.e., the learning/transfer epochs occur later in time than the baseline epochs, and thus any learning/transfer-related structure in the neural data could simply reflect subjects' waning attention or fatigue during MRI testing. To explore this possibility, we tested a fourth model that merely encoded the passage of time ('Time' model in Fig 3D), in which time-adjacent epochs were coded as being the most similar. Note that this RSM model had the lowest performance of all four models, and had significantly less explanatory power than both the 'Learning' and 'Hand' models (Fig 3E). Taken together, these RSAs further support the notion that both effector and learning/transfer-related information are present in the whole-brain manifold data. This prompted us to next consider what specific sets of brain regions are responsible for these different representational structures in the data.

### Regional changes in manifold structure during learning and transfer

In order to explore which particular brain regions exhibited significant changes in manifold eccentricity during the task, we performed, for each brain region, a two-way repeated measures ANOVA (rmANOVA) with hand (2 levels: left, right) and task epoch (3 levels: Baseline, Early, and Late learning) as factors. This approach allowed us to identify brain regions whose connectivity was selectively modulated based on the effector being used (i.e., main effect of 'Hand') versus the specific phase of the task, independent of the effector (i.e., main effect of 'Task Epoch'). We then corrected the results for multiple comparisons using false discovery rate (FDR) corrections (FDR; $q < 0.05$) applied across all brain regions. This two-way rmANOVA revealed that 30 brain regions (26 in cortex, 4 in cerebellum) exhibited a significant main effect of Hand (Fig 4A), whereas 51 regions (47 in cortex, 2 in subcortex, and 2 in cerebellum) exhibited a significant main effect of Task Epoch (Fig 5A). In addition, we found that only four regions exhibited a significant Hand × Task Epoch interaction (detailed in Fig C in S1 Text due to their limited number and straightforward effects, primarily showing the largest eccentricity change during RH Learning Early). When compared to the much larger number of brain regions exhibiting a main effect of Task Epoch (51 regions in total), this suggests that the majority of brain areas displayed similar patterns of manifold eccentricity across both the RH learning and LH transfer epochs (Baseline, Early, and Late). Notably, further analyses on the underlying time series data revealed that these changes in manifold eccentricity did not reflect relative increases versus decreases in the BOLD response across different epochs (see Fig D in S1 Text).

For regions that showed a main effect of Hand, we mainly observed cortical clusters in bilateral somatomotor cortex and posterior cingulate cortex (PCC), as well as a few unilateral areas in visual, premotor, and supplementary motor cortex (Fig 4A). In addition, in the cerebellum we observed significant bilateral clusters in its motor regions (area M4), but right-lateralized effects in the multi-demand (D1–D4) regions [60] (Fig 4A, top). In totality, the majority of these significant brain regions tended to fall within brain networks previously implicated in motor learning and control (e.g., Somatomotor network and the cerebellum, Fig 4B) [7]. To examine the directionality of these hand effects, for each significant brain region we performed a follow-up paired-samples *t* test using the contrast of Right hand > Left hand. This analysis revealed a striking contralateral organization, whereby left hemisphere regions tended to exhibit increased eccentricity when subjects used their right (contralateral) hand during baseline and learning/transfer trials (red areas in Fig 4E) and vice versa for right hemisphere regions (blue areas in Fig 4E). Notably, this same topographic organization was absent in the cerebellum, with all regions exhibiting increased eccentricity when subjects used their dominant right hand (red areas in Fig

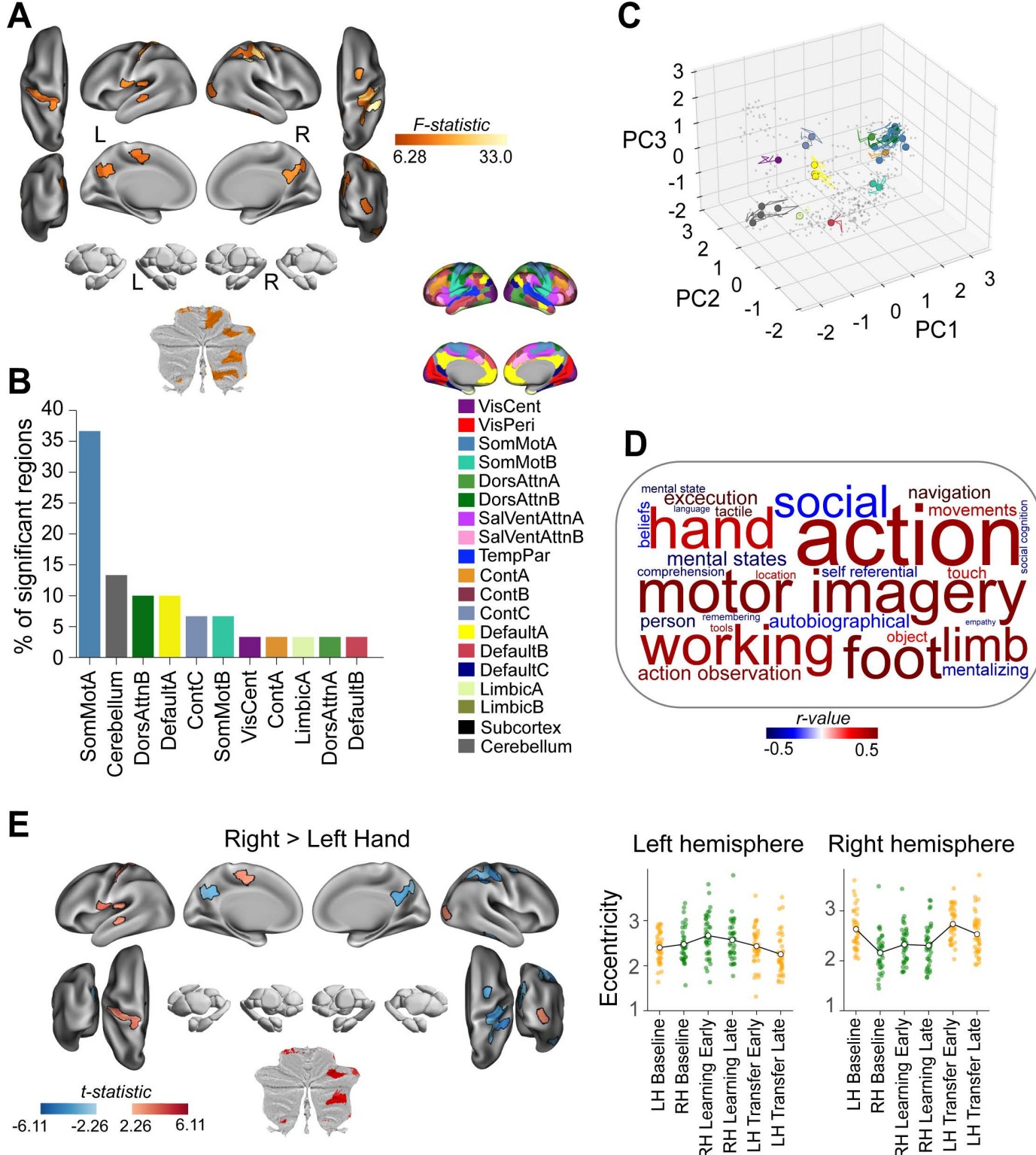

**Fig 4. Effector-related changes in manifold eccentricity. (A)** Brain areas showing a significant main-effect of Hand, based on region-wise two-way repeated measures ANOVAs using a false discovery rate (FDR) correction for multiple comparisons ($q < 0.05$). **(B)** Aggregation of significant brain regions in (A) according to functional network assignment (Yeo 17-networks parcellation [70]). **(C)** Temporal trajectories of significant regions from A in

low-dimensional manifold space. Colored circles indicate each region's initial position during the LH Baseline epoch, and the accompanying trace shows the unfolding displacement of that region across the subsequent five phases of the task (RH Baseline, RH Learning Early, RH Learning Late, LH Transfer Early, and LH Transfer Late). Nonsignificant regions are shown in gray point cloud. **(D)** Meta-analyses of the main effects depicted in A based on the NiMARE correlation decoder tool [75] with the Neurosynth database [76]. Word clouds show the top and bottom 15 processes that are associated with the brain map in **A.** The text size and color within each word cloud denote their correlation value (bigger words have higher correlation values) and their polarity (words in red and blue are positively and negatively associated, respectively, with the main effect brain maps). **(E)** Patterns of regional changes in manifold eccentricity underlying the main effects in A. Left, Pairwise contrast denoting the significant main effect of Hand. Note the general contra-lateral organization of eccentricity changes. Right, Scatter plots show the eccentricity for each significant region (averaged across participants) and separated according to hemisphere (left vs. right). The line plot overlays show the group mean (across ROIs) over task epochs. Note that data points are color-coded, as in Fig 1, according to the hand used during each epoch (orange = left hand; green = right hand). Left = left; Right = right. The data and code needed to generate this figure can be found in https://zenodo.org/records/15648991.

4E). Collectively, these findings suggest that the effector used in the task primarily modulates the manifold embedding of brain regions located closer to the final motor output pathway (e.g., somatomotor cortex and motor cerebellum).

By contrast, for regions that showed a main effect of Task Epoch, we observed major cortical clusters across several regions in higher-order transmodal (association) cortex (Fig 5A). This included bilateral clusters in several areas of the DMN, including medial prefrontal cortex (mPFC), inferior frontal gyrus (IFG), and middle temporal cortex. Notably, we also observed significant effects in bilateral amygdala, and in the association regions of the right cerebellum, including both the demand (area D4) and social-linguistic-spatial (area S2) regions [60] (Fig 5A). In totality, the majority of these significant brain regions tended to fall within brain networks previously implicated in higher-order cognitive functions (i.e., the DMN-B and DMN-A networks, see Fig 5B).

As with the Hand effects, we performed follow-up paired-samples *t* tests to examine changes in the manifold eccentricity of these regions across task epochs. Firstly, to examine how the eccentricity of these regions changed at the onset of both RH learning and LH transfer trials, we performed a contrast of Early > Baseline (averaged across left- and right-hand trials; Fig 5E, top). This contrast revealed that nearly every brain region showing a main effect of Task Epoch exhibited significant manifold expansion (i.e., increased eccentricity) from Baseline to Early learning/transfer, suggesting that these regions became more segregated from other brain networks during the Early learning/transfer phases (red areas in Fig 5E, top). Secondly, to examine how eccentricity changed over the course of learning, we performed a direct contrast of Late > Early learning/transfer (again, averaged across left-and right-hand trials; Fig 5E, bottom). Strikingly, this contrast revealed a near complete reversal in the general pattern of effects noted above, which the vast majority of the DMN regions, as well as the right amygdala, now exhibiting significant contraction along the manifold (i.e., a decrease in eccentricity; blue areas in Fig 5E, bottom), suggesting their increased integration with other brain networks.

For completeness, and to provide a comparison with these FC-derived results, we also performed a standard mass-univariate General Linear Model (GLM) analysis to identify brain regions showing significant changes in mean BOLD activation across the task epochs (see Materials and methods). The results of this analysis (Fig E in S1 Text) revealed a distinct pattern of activation changes across epochs, which, while partly consistent with prior activation-based studies of visuomotor adaptation [8,77,78], did not closely correspond to the DMN-centric connectivity reorganization highlighted by our manifold analyses above. This divergence underscores that GLM activation and FC capture fundamentally different, albeit complementary aspects of neural activity underlying learning and transfer.

## Meta-analysis of regional changes in manifold embedding

To probe the putative behavioral-cognitive processes supported by the distinct patterns of brain activity identified above, we performed a meta-analysis on the regional eccentricity maps in Figs 4A and 5A using the Neurosynth database [76] and NiMARE library [75]. This meta-analytic decoding approach allows us to interpret the functional significance of our neural findings by linking the observed patterns of brain activity to cognitive processes identified in the broader neuroimaging literature. By doing so, we can gain insights into the potential behavioral and cognitive functions associated with

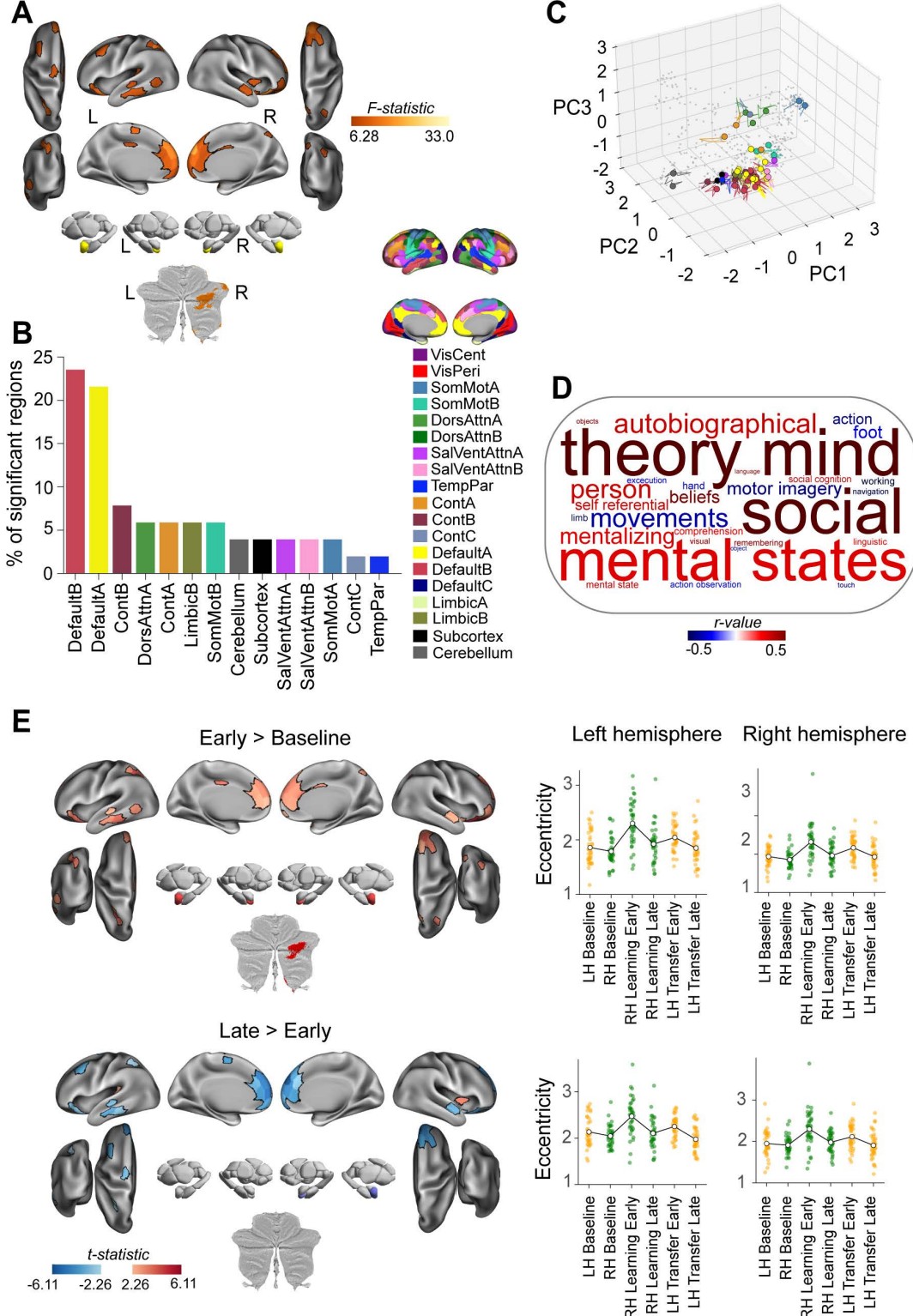

**Fig 5. Learning/transfer-related changes in manifold eccentricity. (A)** Brain areas showing a significant main-effect of Task epoch, based on region-wise two-way repeated measures ANOVAs using a false discovery rate (FDR) correction for multiple comparisons ($q < 0.05$). **(B)** Aggregation of significant brain regions in (A) according to functional network assignment (Yeo 17-networks parcellation [70]). **(C)** Temporal trajectories of significant regions

from (A) in low-dimensional manifold space (plotted the same as in Fig 4). **(D)** Meta-analyses of the main effects depicted in A based on the NiMARE correlation decoder tool [75] with the Neurosynth database [76]. See Fig 4 caption for details. **(E)** Patterns of regional changes in manifold eccentricity underlying the main effects in A. Pairwise contrasts (left) and eccentricity plots (right) denoting the significant main effect of Task epoch. The eccentricity plots highlight the noticeable increases in eccentricity during each of the RH Learning Early and LH Transfer Early epochs. Left = left; Right = right. The data and code needed to generate this figure can be found in https://zenodo.org/records/15648991.

the regions showing significant changes in eccentricity, thereby enhancing our understanding of the underlying processes supporting motor learning and generalization.

The results of this meta-analytic decoding analysis are depicted in the form of word clouds in Figs 4D and 5D. In these visualizations, keywords/descriptors associated with whole-brain patterns of brain activity from other fMRI studies that closely resemble our own brain maps receive a higher weighting (and thus a larger font size), with the polarity (blueness or redness of the text) describing the direction of this resemblance. For instance, from the fMRI literature we find that our main effect of Hand F-stat brain map (in Fig 4A) is positively correlated with functional terms like 'action', 'hand' and 'motor imagery', and negatively correlated with terms like 'social', 'mental states' and 'autobiographical' (Fig 4D). This result validates our findings by confirming that regions associated with hand movements and motor control are indeed implicated when participants use different effectors.

By contrast, for our main effect of Task Epoch F-stat map (in Fig 5A), we find that this brain map is positively correlated with terms like 'social' and 'mental states', as well as several other higher-order cognitive processes (e.g., 'theory of mind'); whereas it is negatively correlated with terms like 'movements', 'motor imagery' and 'action' (Fig 5D). This suggests that the regions involved in learning and generalization are linked to higher-order cognitive functions rather than traditional motor functions. At first glance, this outcome may seem counterintuitive given that the Task Epoch F-stat map describes neural changes associated with *motor* learning and generalization. However, it is generally consistent with contemporary work positing that explicit cognitive processes, which underpin many abstract higher-level functions (e.g., theory of mind), also play a critical role in motor learning and generalization [30,34,79]. Thus, these analyses not only corroborate our findings by associating motor-related brain maps with motor functions but also reveal that the regions involved in learning and generalization are linked to higher-order cognitive processes.

### Evidence that the manifold patterns during early learning are re-expressed during transfer

Accompanying the brain map results in Figs 4E and 5E are scatterplots (rightmost panels) that denote the eccentricity changes of each significant brain region across epochs for each pairwise contrast. In the case of the brain regions showing a main effect of Hand (shown in Fig 4A), these eccentricity plots clearly delineate the strong contralateral effects described above (and shown in Fig 4E); i.e., the eccentricity of left hemisphere regions is higher, on average, for RH trials, and vice versa for the right hemisphere regions on LH trials. However, in the case of the brain regions showing a main effect of Task Epoch (shown in Fig 5A), there are two important observations to be made. Firstly, unlike the contralateral hand effects (in Fig 4E), the eccentricity changes in both hemispheres are highly similar (compare left and right hemisphere eccentricity plots in Fig 5E). This indicates that the connectivity changes occurring across the baseline, early, and late learning/transfer epochs are largely bilateral in nature, and mainly localized to higher-order, non-motor regions of cortex. Secondly, there is a particularly large increase in the manifold eccentricity of these regions during the first early learning period, which is followed by a second (more moderate) increase in eccentricity during the early transfer period (see Fig 5E). This second increase in eccentricity, along with the resulting main effect, suggests that the manifold structure observed during initial RH learning may be selectively 're-expressed' during initial LH transfer.

To explore this idea directly, moving beyond simple magnitude comparisons, we used an RSA approach to test whether the spatial pattern of regional eccentricity was similar between the two epochs. Specifically, we tested if the spatial correlation between the pattern of regional eccentricity during the RH Learning Early and the LH Transfer Early epochs is

significantly higher than the correlation between the RH Learning Early epoch and the other learning epochs that bookend the LH Transfer Early epoch (i.e., RH Learning Late and LH Transfer Late epochs). This RSA analysis employs a logic similar to that used to measure cortical reinstatement in episodic memory research, where neural activity patterns during encoding and retrieval are directly compared [80–89]. When we performed this analysis, we found that the correlation between the RH Learning Early epoch and each of the three subsequent epochs significantly differed (rmANOVA, $F(2,74)$ = 3.14, $p = 0.049$). Follow-up paired-samples $t$ tests revealed that the spatial correlation between the RH Learning Early epoch and the LH Transfer Early epoch was indeed significantly higher than compared to the RH Learning Early epochs' correlation with each of RH Learning Late (one-tailed $t$ test, $t(37) = 1.87$, $p = 0.035$) and LH late learning (one-tailed $t$ test, $t(37) = 2.58$, $p = 0.007$; Fig 6B) epochs. Together, these findings provide compelling evidence that the manifold structure established during RH Learning Early is most strongly re-expressed during the early phase of transfer to the untrained left hand.

Note that one alternative explanation of these above results could be that these 're-expression' effects are not specific to higher-order cortical areas per se, but rather reflect a brain-wide phenomenon. To explore this possibility, we performed the exact same RSA on brain regions exhibiting a main effect of Hand (in Fig 4A). We reasoned that these brain areas, primarily localized to sensorimotor cortex and which predominantly showed contralateral effects (Fig 4E), should demonstrate hand-specific patterns distinct from the re-expression effect observed in the higher-order regions, such as the DMN. Specifically, we would predict that the correlation between the pattern of regional eccentricity during the RH Learning Early and RH Learning Late epochs should be significantly higher than the correlation between the former condition and the two LH transfer epochs (early and late). When we performed this RSA on the brain regions exhibiting a main effect of Hand, we again found that the correlation between the RH Learning Early epoch and each of the three subsequent epochs

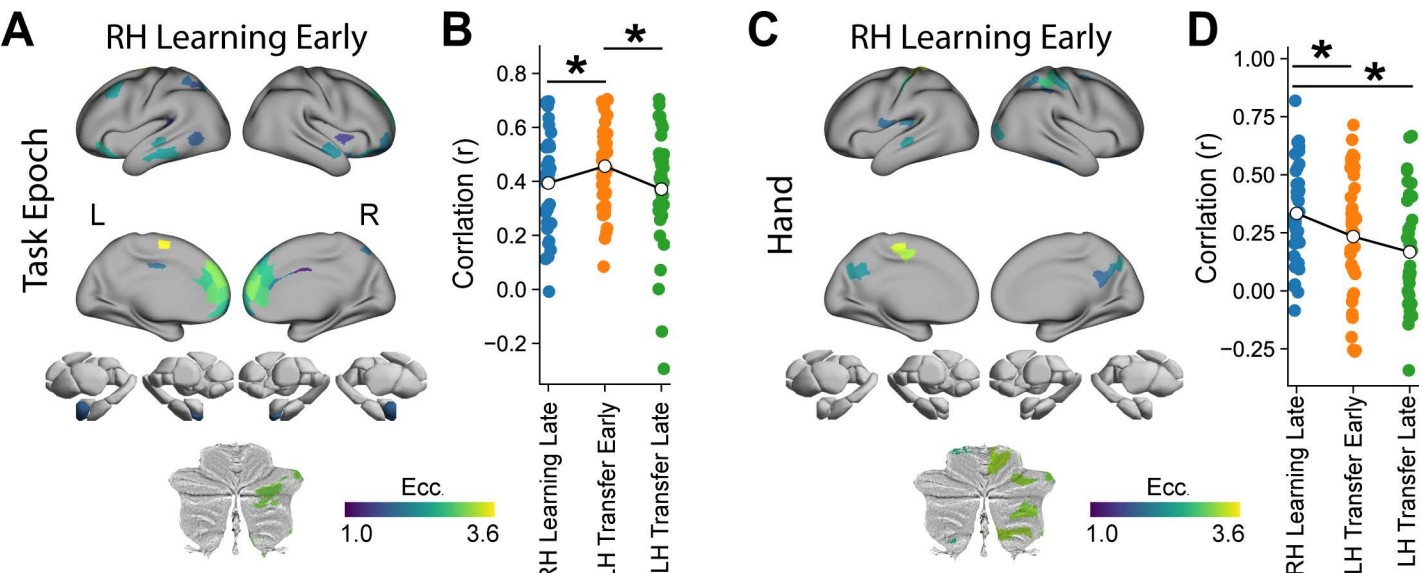

**Fig 6. Re-expression of early learning manifold structure across epochs. (A)** Mean eccentricity values for the RH Learning Early epoch for brain regions exhibiting a significant main effect of Task Epoch (i.e., areas from Fig 5A). **(B)** Pattern similarity (Pearson r) between RH Learning Early and the three subsequent task epochs—RH Learning Late, LH Transfer Early, and LH Transfer Late—computed across the regions shown in A. The black line shows the across-subject mean, and individual points represent single subjects. **(C)** Mean eccentricity values for the RH Learning Early epoch for brain regions exhibiting a significant main effect of Hand (i.e., areas from Fig 4A). **(D)** Pattern similarity between RH Learning Early and the same three comparison epochs as in (A), but computed across the Hand-significant regions shown in C. * denotes $p < 0.05$ for one-tailed paired-samples $t$ tests. The data and code needed to generate this figure can be found in https://zenodo.org/records/15648991.

significantly differed (rmANOVA, $F(2,74) = 5.08$, $p = 0.008$). Consistent with our predictions, follow-up paired-samples $t$ tests revealed that the highest correlation with the RH Learning Early epoch was for the RH Learning Late epoch, and that the correlations were subsequently lower for both the LH early (one-tailed $t$ test, $t(37) = −1.84$, $p = 0.037$) and late (one-tailed $t$ test, $t(37) = −3.10$, $p = 0.002$) transfer epochs (see Fig 6D). This contrasting hand-specific pattern in sensorimotor regions serves as a valuable control, as it further highlights the selective nature of the re-expression effects observed in the higher-order cortical areas, such as the DMN (Fig 6B).

## Changes in regional manifold embedding relate to underlying brain structure

A unique structural property of DMN brain areas is that, compared to other regions (e.g., sensory-motor cortex), they tend to exhibit lower myelin content. As a consequence, they also show greater complexity in their patterns of dendritic branching and arborization [90–92]. These structural characteristics have been hypothesized to facilitate the integrative capacity of brain regions and their potential for both learning [93,94] and contextual-processing [95,96]. Similarly, DMN areas have been shown to exhibit a higher density of neurotransmitter receptors (e.g., serotonin, dopamine, etc.) as compared to several other cortical areas [52,53], which is hypothesized to increase their capacity for cognitive flexibility. Given these neuroanatomical observations, we naturally wondered whether our Hand and Task Epoch effects might relate to differences in the distribution of both myelin content and receptor density across cortex. To examine this, we built null models through spin permutation testing procedures [97,98] that account for the spatial autocorrelation in our two F-stat brain maps (Hand and Task Epoch) and tested for their spatial correlation with (1) the Human Connectome Project 1200 (HPC) myelin map (T1w/T2w ratio) [98] and (2) the first principal gradient (PC1) of receptor density from Hansen and colleagues [52], which captures total neurotransmitter receptor density per brain area. This permutation 'spin-testing' approach revealed that brain areas selectively modulated by the effector (Hand) were associated with *higher* myelin content ($r = 0.24$, $P_{spin} = 0.013$) and *lower* receptor density ($r = −0.14$, $P_{spin} = 0.048$), whereas brain areas selectively modulated by the learning/transfer phases of the task tended were associated with *lower* myelin content ($r = −0.27$, $P_{spin} = 0.039$) and *higher* receptor density ($r = 0.20$, $P_{spin} = 0.024$; see Fig 7). These results imply that differences in the localization of our two main effects may result, in part, from the underlying structural features (i.e., myelination and receptor density) of those different brain areas.

## Alterations in connectivity that underlie changes in manifold structure

While significant changes in the manifold embedding of individual brain regions indicate an alteration in those regions' patterns of whole-brain connectivity, the precise nature of these connectivity alterations is not obvious. To examine this, and to better elucidate the nature of the Hand and Task Epoch-related main effects observed above, we performed seed connectivity analyses using representative regions from the left and right hemispheres of two different brain regions: (1) Primary motor cortex (M1), which exhibited a main effect of Hand in our analyses, and (2) medial mPFC, which exhibited a main effect of Task Epoch. Together, the selection of these seed regions allows us to characterize representative changes in the patterns of connectivity across lower-level sensory-motor regions (M1) versus higher-order transmodal regions.

For each seed region, we contrasted the associated whole-brain connectivity maps for specific pairwise comparisons that allowed us to elucidate the key main effects. For the M1 seed regions, this involved averaging the seed connectivity maps based on the effector used (left versus right hand) and then performing region-wise paired-samples $t$ tests for the contrast of Contralateral > Ipsilateral hand (note that, with this contrast, the direction of effect is based on the seeded hemisphere, thus aiding the interpretation of the maps). For the MFPC seed regions, this involved performing two sets of contrasts that allowed us to capture learning-related effects over time: (1) the change from Baseline to Early Learning/Transfer (Early > Baseline), and (2) the change from Early to Late Learning/Transfer (Late > Early). The results of these different seed-based contrasts are shown in Fig 8, where we present the unthresholded region-wise contrast maps to provide a comprehensive, multivariate view of the region's change in whole-brain connectivity.

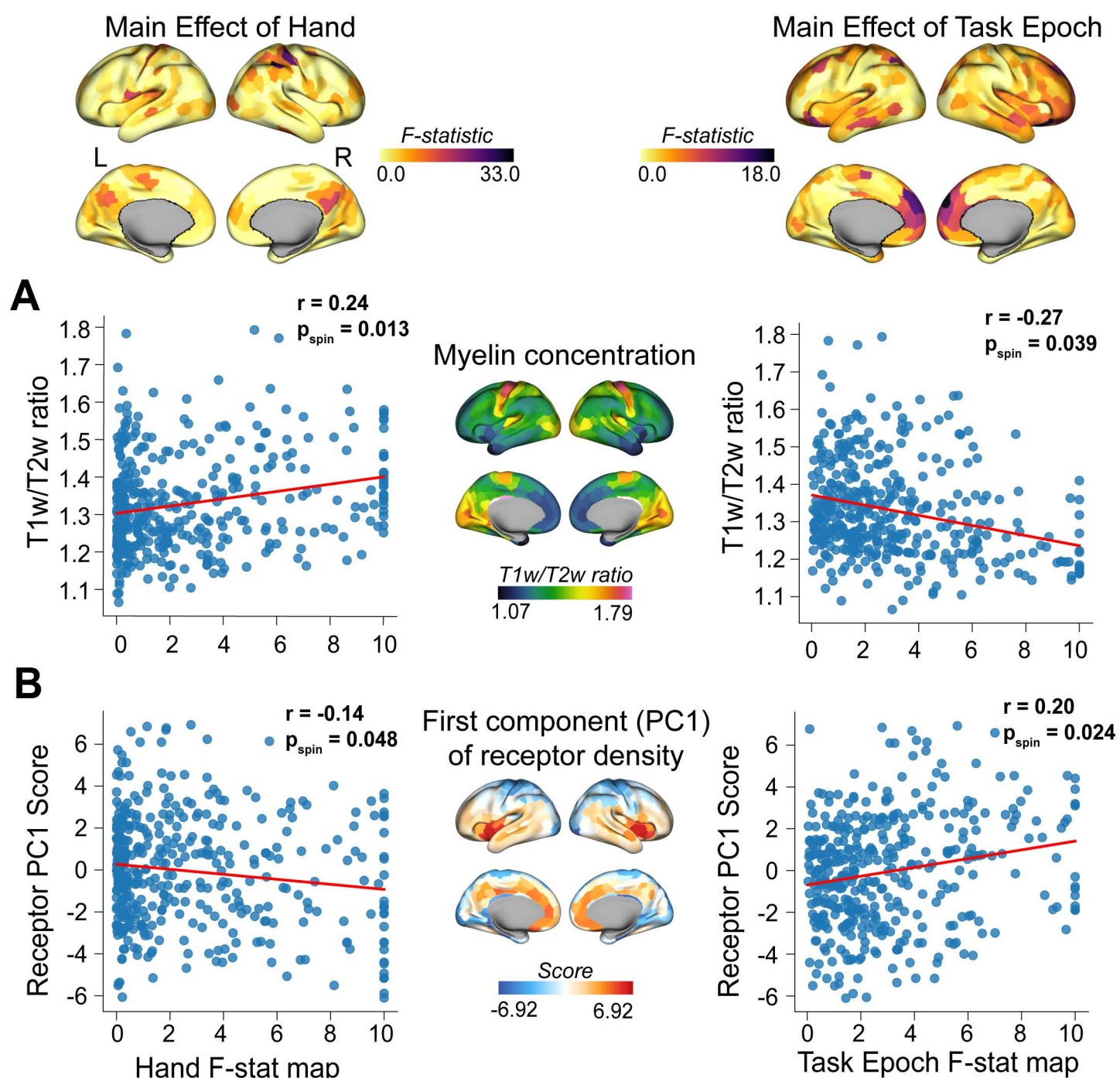

**Fig 7. Effector-related and learning/transfer-related main-effects relate to underlying brain structure.** Spatial correlations between our two main-effect F-stat maps (shown at top) with **(A)** cortical myelin concentration (T1w/T2w ratio) and **(B)** the first principal component of receptor density (B, from [52]). Spatial null model testing was performed using the Neuromaps toolbox [98,97]. L=left; R=right.

For the left M1 seed region, we found that use of the contralateral (dominant) right hand resulted in relatively constrained patches of increased bilateral connectivity with the adjacent sensorimotor cortex (i.e., premotor, motor, supplementary motor, and anterior parietal cortex), as well as increased connectivity with bilateral motor areas of the cerebellum (Fig 8A). By contrast, for the right M1 seed region, we found that use of the contralateral (non-dominant) left hand resulted in increased connectivity with much larger swaths of bilateral motor, premotor, and supplementary motor cortex, as well as large swaths of visual cortex and parietal cortex. In addition, we observed increased bilateral connectivity with medial temporal aspects of subcortex and association regions of the cerebellum (Fig 8B). This general observation conforms with recent work in right-handers showing that left (non-dominant), compared to right (dominant), hand movements are associated with more extensive interhemispheric connectivity [99] and overall brain activity [100]. Together, these results suggest that, generally speaking, the manifold expansions of left and right M1 when using the contralateral hand mainly arise from their increased connectivity with other sensory-motor areas in cortex. This highlights how increased connectivity within a functional system can lead to greater functional segregation of that system relative to the whole-brain average, manifesting as increased manifold eccentricity.

For the left MPFC seed region of the DMN, we found that, during early learning/transfer (Early > Baseline), the MPFC exhibited increased connectivity with other bilateral DMN subregions (e.g., several areas in MPFC, angular gyrus, IFG) and networks in association cortex (e.g., Limbic and SalVentAttn networks), large swaths of subcortex, and association areas of left cerebellum. In addition, early learning/transfer was marked by general decreases in MPFC connectivity with sensory-motor areas (e.g., visual and somatomotor cortex) and motor areas of the right cerebellum. By contrast, during late learning (Late > Early), we observed a strong reversal in these patterns of connectivity, with the MPFC area now exhibiting increased connectivity with areas of sensory-motor cortex (e.g., visual cortex, somatomotor cortex, premotor and anterior parietal cortex) and motor cerebellum, as well as generally reduced connectivity with other DMN areas, and association areas of the cerebellum (Fig 8C). Notably, for the right MPFC seed region, we observed a strikingly similar pattern of changes in whole-brain connectivity from both Baseline to Early learning/transfer and from Early to Late learning/transfer (consistent with the main effects being bilateral in nature; Fig 8D). Together, these results suggest that the manifold expansion of the MPFC area during early learning/transfer mainly emerges due to its increased connectivity with other transmodal brain areas (i.e., segregation of transmodal cortex) whereas the manifold contraction of this region during late learning/transfer mainly emerges due to its increased connectivity with non-transmodal cortical areas, such as somatomotor cortex.

## Changes in whole-brain manifold structure relate to performance

Up to this point, we have characterized the mean, group-level changes in manifold structure across the learning and transfer phases of the task. Yet, it is well known that subjects can differ markedly in their motor learning abilities [33,35,101,102]. Indeed, while the group-averaged learning curve in Fig 1B shows that subjects, on average, were able to reduce their visuomotor errors during the task, such learning curves tend to mask significant intersubject variability (Fig 9A and 9B). To help capture the inter-subject variability in transfer performance, and in line with more traditional measures of intermanual transfer [27,55,79,103], we also calculated a 'transfer rate' measure for each participant, defined as the difference between their initial error in the left hand transfer block (LH Transfer Early Error) and their final error in the right hand learning block (RH Learning Late Error). This transfer rate measure reflects the degree to which learning generalizes across hands, with lower transfer error indicating better generalization (Fig 9B). Perhaps unsurprisingly, we observed a very strong positive linear relationship between subjects' LH Transfer Early Error and this Transfer Rate measure ($r = 0.73$; Fig 9D).

Prior behavioral work, including our own [33,35,36,55], has shown that subjects vary greatly in their use of explicit cognitive re-aiming strategies during learning, and that subject learning performance is related to differences in strategy use. In addition, it has been shown that cognitive strategies developed during sensorimotor adaptation with one hand can

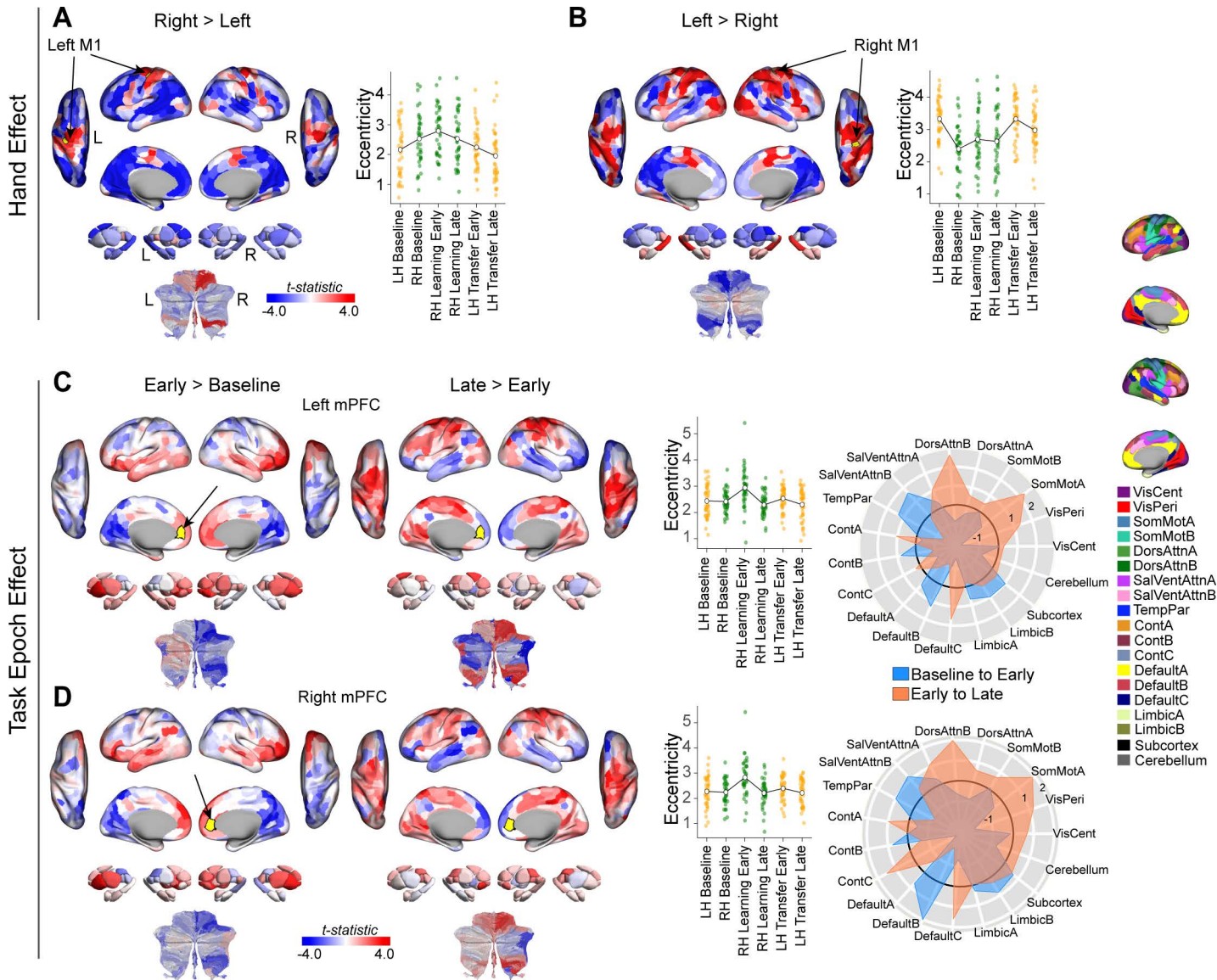

**Fig 8. Connectivity changes that underlie the main effects of hand and task epoch. (A and B)** Connectivity changes for each M1 seed region. Seed regions are denoted in yellow and indicated by arrows. Positive (red) and negative (blue) values show increases and decreases in connectivity for the contrast of Contralateral > Ipsilateral hand. Scatter plots at right show the eccentricity for each participant, and the line plot overlays show the group mean across task epochs. Note that data points are color-coded, as in Fig 1, according to the hand used during each epoch (orange = left hand; green = right hand). **(C and D)** Connectivity changes for each MPFC seed region (denoted in yellow). Positive (red) and negative (blue) values show increases and decreases in connectivity, respectively, for Baseline to Early learning/transfer (leftmost panel) and for Early to Late learning/transfer (adjacent right panel). Eccentricity scatter plots (middle) are the same as in A and B. Rightmost panel contains spider (polar) plots, which summarize these patterns of changes in connectivity at the network-level (according to the Yeo 17-networks parcellation [70]). Note that the black circle in the spider plot denotes $t = 0$ (i.e., zero change in eccentricity between the epochs being compared). The data and code needed to generate this figure can be found in https://zenodo.org/records/15648991.

be readily transferred to the untrained hand [27,28,79]. Consistent with these prior observations, we observed significant individual variability in subjects' reported aiming directions, with some subjects reporting zero explicit knowledge of the rotation (i.e., they aimed directly at the target, or ~0° explicit report) compared to other subjects who appeared to exhibit

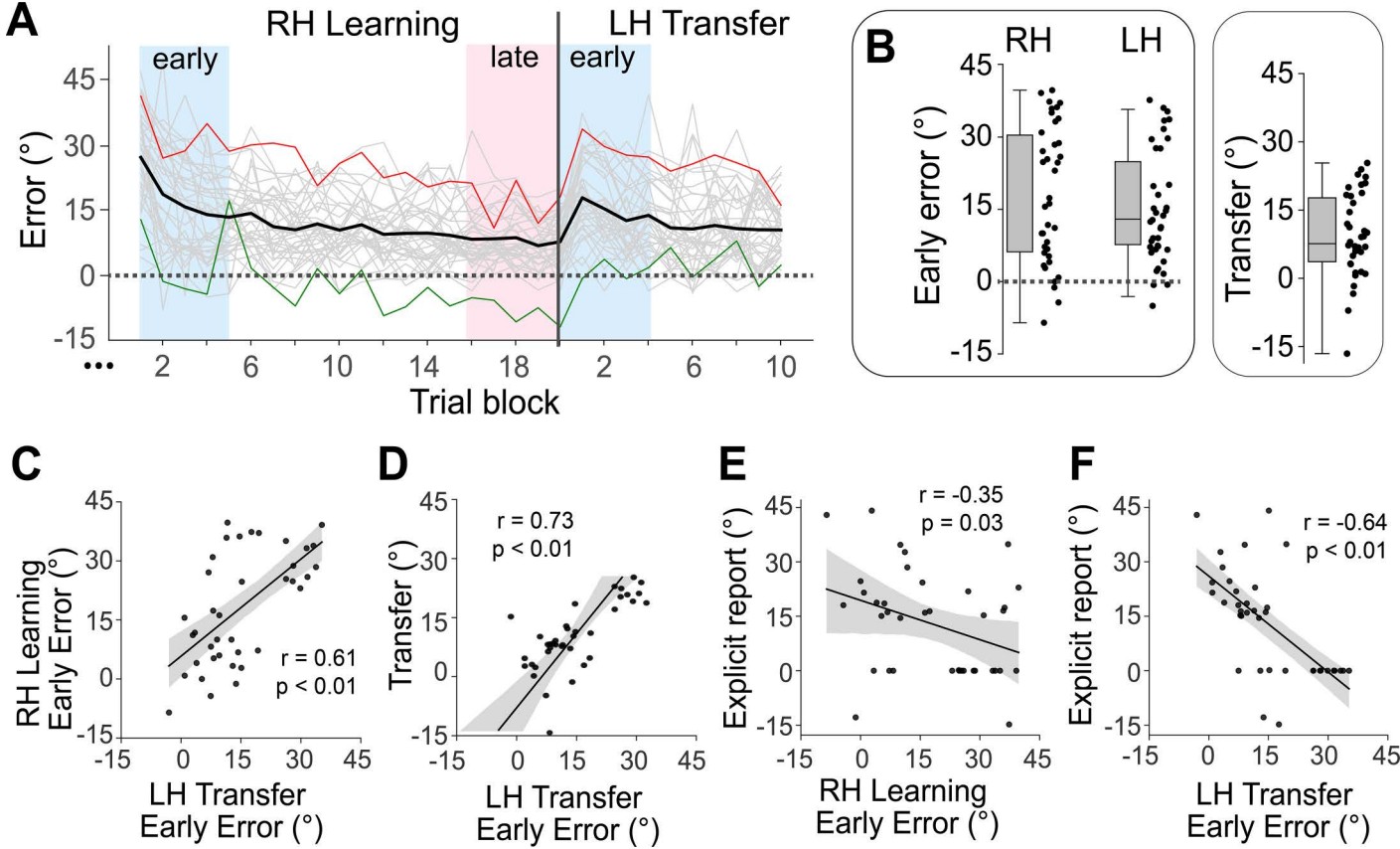

**Fig 9. Individual differences in motor learning and generalization. (A)** Individual subject learning curves. The Solid black line denotes the mean across all subjects, binned by trial block, whereas the light gray traces denote individual participants. The green and red traces denote the learning curves of a good (low error) and poor (high error) learner, respectively. **(B)** Distribution of median angular error during the RH Learning Early and LH Transfer Early epochs (corresponding to the timepoints covered by the faint blue boxes in A), as well as Transfer Rate, constructed by subtracting the RH learning Late angular error (red box) from the LH Transfer Early epoch error. Single data points denote the median error of individuals. **(C)** Correlation between subjects' angular error during the RH Learning Early and LH Transfer Early epochs. **(D)** Correlation between subjects' Transfer Rate and angular error during the LH Transfer Early epoch. **(E)** Correlation between subjects' explicit reports (collected at the end of the Learning block) and angular error during the RH Learning Early epoch **(F)** Same as E, but for the angular error during the LH Transfer Early epoch. Single data points denote individuals, and the black line denotes the best-fit regression line, with shading indicating ±1 standard error of the mean (SEM). The data and code needed to generate this figure can be found in https://zenodo.org/records/15648991.

near complete compensation for the VMR (i.e., ~45° explicit report; see scatterplots in Fig 9E and 9F). In addition, we found that subjects' median angular errors during each of the RH Learning Early and LH Transfer Early periods were negatively correlated to their reported re-aiming strategy ($r = -0.35$ and $r = -0.64$, respectively; see Fig 9E and 9F), and that subjects' errors during each of these two periods were highly positively correlated ($r = 0.61$, Fig 9C). Together, this indicates that subjects' who reported using a re-aiming strategy tended to learn more quickly at the task [see also 33,35], and that this re-aiming strategy transfers across the hands [see also 56,82], respectively. Given the existence of these behavioral relationships, we wondered whether there also existed correlations between individual subjects' performance and the corresponding changes in manifold structure during these early RH learning and LH transfer periods.

To test this, for each brain region and each of the RH Learning Early and LH Transfer Early epochs, we calculated the correlation between subjects' median angular error and the corresponding change in eccentricity from the Baseline to Early epochs for each hand (i.e., RH Learning Early > RH Baseline and LH Transfer Early > LH Baseline, Fig 10A and

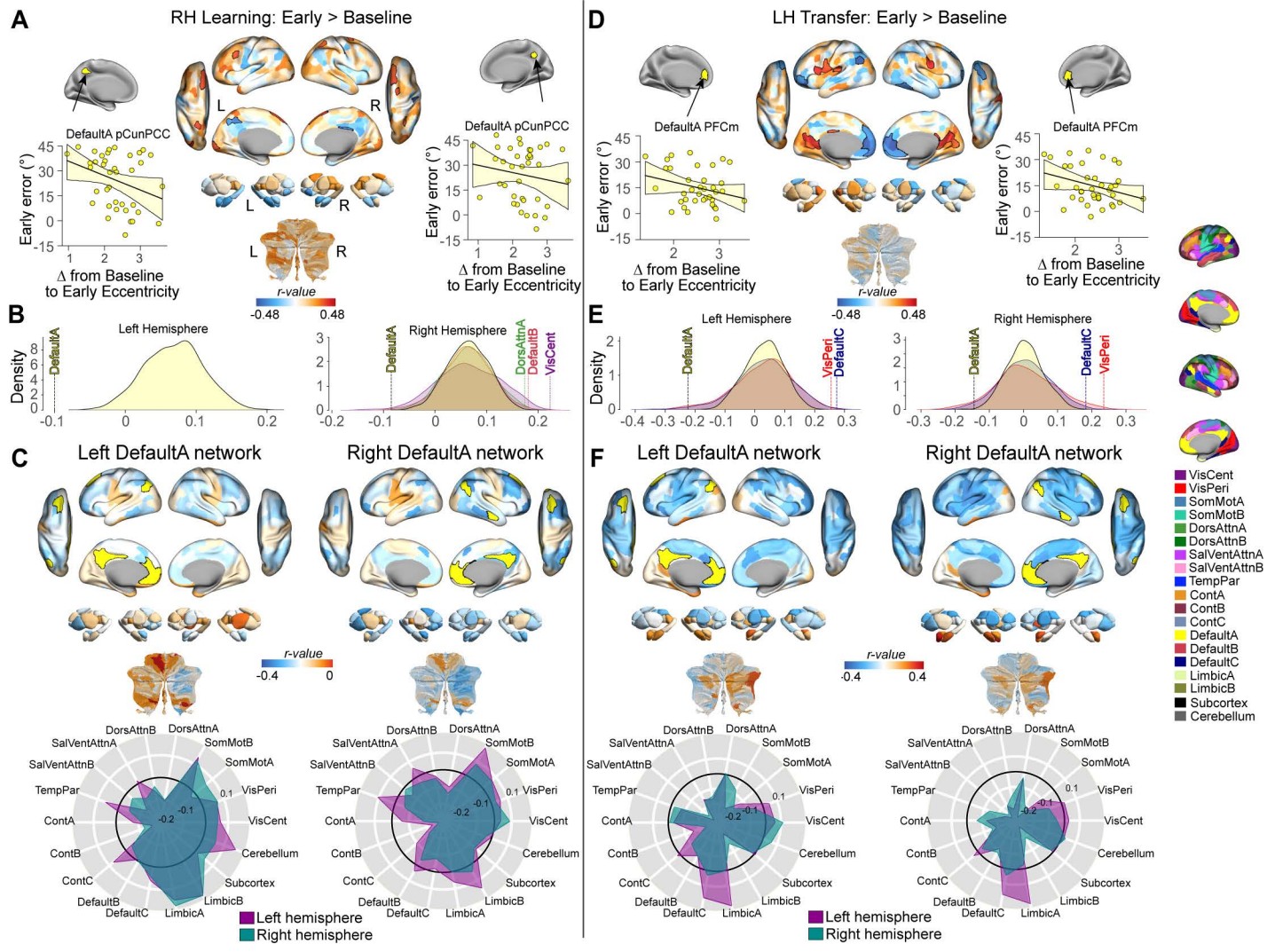

**Fig 10. Relationship between learning performance and learning-related changes in eccentricity. (A)** Whole-brain correlation map between subjects' RH early error and the change in regional eccentricity from RH Baseline to RH Learning Early. Black bordering denotes regions that are significant at *p* < 0.05. **(B)** Results of the spin-test permutation procedure, assessing whether the spatial topography of correlations in A are specific to individual functional brain networks in each hemisphere. The density graph denotes the null distribution for statistically significant brain networks only, as derived from 1,000 iterations of a spatial autocorrelation-preserving null model [98,97] (see Fig G in S1 Text for the results from other non-significant networks). The dashed vertical line denotes the true correlation value. All correlations were corrected for multiple comparisons (*q* < 0.05). Scatterplots above panel B show the correlation between the change in eccentricity for a representative brain region from each significant network (region denoted in yellow) with subjects' median angular error during the RH Learning Early epoch. **(C)** Underlying pattern of functional network connectivity, and its relationship to learning performance, for two of the significant networks in B. Positive (red) and negative (blue) values denote where an increase in inter-network connectivity was associated with either higher or lower angular errors, respectively (i.e., blue values denote where increased connectivity led to lower errors, or better performance). Spider plots, at bottom, summarize these patterns of correlation changes at the network-level. Note that the black circle in the spider plot denotes *r* = 0 (i.e., zero correlation between the change in functional connectivity and performance). **(D)** Same as A but for the correlation between subjects' LH early error and the change in regional eccentricity from LH Baseline to LH Transfer Early. **(E)** Same as B, but for the correlation map in D. **(F)** Underlying pattern of functional network connectivity, and its relationship to transfer performance, for two of the significant networks in E (see Fig G in S1 Text for the results from other non-significant networks). Data is presented the same as in C, but for angular errors during the LH Transfer Early epoch. Again, here, blue values denote where increased connectivity was associated with lower errors.

10D, respectively). [Note that for an analysis of the correlation between the eccentricity change from LH Baseline to LH Transfer Early versus transfer rate, please see Fig F in S1 Text]. Crucially, because individual differences in angular error during these phases are strongly linked to the reported use of explicit strategies (as established in Fig 9E and 9F), these brain-behavior correlations also allow us to probe, at least in part, the neural dynamics underlying the engagement of explicit cognitive processes during adaptation and transfer. After applying FDR-corrections to account for multiple comparisons ($q < 0.05$), this analysis did not identify any significant regional correlations for either the RH Learning Early or LH Transfer Early epochs. However, it is important to consider that these corrections for multiple comparison testing do not take into account the spatial topography of effects across brain regions [97,98], which may correspond to known functional networks. Indeed, visual inspection of the whole-brain correlation map for the LH Transfer Early epoch indicates a high degree of spatial contiguity in the significant effects (areas bordered in black, $p < 0.05$), particularly within areas of higher-order transmodal cortex, such as MFPC and IFG (Fig 10D). This spatial topography strongly implies that any association between eccentricity and subject learning performance may be better captured at the level of distributed whole-brain functional networks.

To examine the relationship between learning performance and changes in manifold embedding at the level of distributed functional networks, we employed the Yeo and colleagues 17-network parcellation [70] to map each region onto its corresponding functional network assignment for each hemisphere separately (thus accounting for any potential hemispheric effects). For each participant, we computed the mean manifold eccentricity across regions within each network (network eccentricity) and examined the correlation between the change in eccentricity (from Baseline to Early) of each brain network with subjects' corresponding performance (median angular error). To assess the statistical significance of these network-level correlations, we used the 'spin-test' permutation procedure ([98,97]; see Materials and methods) and corrected for multiple comparisons across all networks using an FDR correction ($q < 0.05$). This analysis revealed that, for the RH Learning Early epoch, changes in eccentricity of the bilateral DMN-A networks, as well as the ipsilateral (right) DMN-B, DAN-A, and VisCent networks, significantly correlated with participants' learning performance (Fig 10B). Specifically, for the DMN-A networks (which mainly encompass mPFC, PCC, and angular gyrus), greater expansion along the manifold during early learning corresponded to lower error (i.e., improved performance). Note that this correlation effect extrapolates upon the group-level Task Epoch effects reported in Fig 4B, indicating that participants who exhibited this group-level effect more prominently tended to exhibit better early learning performance. Conversely, for the DMN-B, DAN-A, and VisCent networks, less expansion along the manifold during early learning was associated with lower error (i.e., better performance).

To further elucidate the underlying connectivity changes driving these network-level eccentricity effects, we examined the relationship between subjects' learning performance and the learning-related change in FC between each of these aforementioned brain networks and every other functional network in the brain (as well as with individual regions in subcortex and cerebellum, which do not have these same functional network assignments). Given that eccentricity serves as a multivariate index of whole-brain changes in connectivity, we have opted, as in the seed-based results in Fig 8, to show the whole-brain, unthresholded correlation maps depicting these various relationships (see Fig 10C and Fig G in S1 Text). For the bilateral DMN-A networks, Fig 10C shows that *increased* FC with the Control-B and SalVentAttn-B networks tended to be associated with lower visuomotor errors (i.e., better performance; these negative correlation relationships are denoted by blue colors on the brain maps). Conversely, we also found that better (worse) performance tended to be associated with *decreased* (increased) FC between the DMN-A and sensory and motor areas, as well as motor areas of the cerebellum (denoted by red colors in the brain maps) (see Fig G in S1 Text for interpretations of the other network effects). The concurrent relative strengthening of DMN-A connectivity with cognitive networks (e.g., Control-B and SalVentAttn-B), which are located at similar positions along the whole-brain manifold (Fig 2C), versus relative weakening of DMN-A connectivity with sensorimotor networks (visual, somatomotor regions), which are located at opposing ends of the whole-brain manifold (Fig 2C), likely explains our observed eccentricity result—namely, the relative increase in the DMN-A's

eccentricity during early learning (see also Fig 8C and 8D). Moreover, our brain-behavior correlation findings further suggest that the relative magnitude of this concurrent strengthening/weakening FC pattern relates to subject performance.

Notably, when we applied the same analyses to the LH Transfer Early phase of the task, we observed a consistent pattern of effects for the DMN-A network. Specifically, we found that an increase in the eccentricity of bilateral DMN-A networks correlated with better performance during transfer trials (Fig 10E). Additionally, we found significant correlations in other DMN (i.e., DMN-C) and Visual (VisPeri) networks, consistent with the same directionality observed during RH Learning Early; i.e., less expansion of these networks along the manifold during the early transfer period corresponded to lower error. Further examination of the underlying changes in between-network connectivity associated with the DMN-A network again revealed that enhanced FC with Control-B and SalVentAttn-B networks was associated with reduced visuomotor errors (Fig 10F). In addition, we also observed that increased connectivity between the DMN-A and each of the somatomotor and limbic-B networks was associated with lower visuomotor errors. Note that when we performed all the same analyses above but instead using subjects' Transfer Rate as our behavioral measure, the results were qualitatively similar—in particular, re-identifying the unique changes in the DMN-A network and its relationship to performance (see Fig F in S1 Text). This is to be expected given the strong positive correlation between subjects' Transfer Rate and their LH Transfer Early Error measure (Fig 9D). Taken together, the consistency of the DMN-A correlation results across both the learning and transfer phases of the task, as well as across both hemispheres, suggest a key role for this particular network in subject-level performance.

## Discussion

The ability to flexibly transfer acquired motor behaviors to new scenarios is a hallmark of successful learning. Yet, how this generalization is achieved by the brain remains obscure. To examine this question, we used functional MRI to investigate the neural bases underlying the transfer of visuomotor adaptation from the trained to untrained hand. Our findings reveal that the initial transfer phase of learning re-expresses the same patterns of brain activity that are observed during the initial adaptation phase. Notably, we found that these changes predominantly involved alterations in the connectivity of DMN brain areas, as opposed to changes throughout the sensorimotor system. Further analyses revealed that these effects were correlated to the unique structural properties of transmodal cortex (its lower myelin content and higher receptor density), and that inter-individual differences in the manifold embedding of DMN areas were related to both learning and generalization performance. Collectively, our results provide a macroscale, whole-brain view of the connectivity changes associated with sensorimotor adaptation and generalization, and offer a new perspective on the DMN's contribution to motor control.

Previous fMRI studies investigating the neural bases of motor generalization have predominantly focused on motor sequence learning, examining the transfer of learned finger movement sequences across the hands (for reviews, see [104–108]). In contrast, research on the neural bases supporting the generalization of sensorimotor adaptation, which involves the transfer of learned action elements rather than their sequence [28,109,110], has been relatively scarce. Of the limited studies available, they have primarily focused on characterizing learning-related increases versus decreases in the activity of individual brain regions, such as visual, parietal, motor, and premotor cortical areas, as well as the cerebellum and striatum [1,7,8,77,78,111]. For instance, fMRI activation studies have highlighted the differential recruitment of these areas depending on whether the dominant or non-dominant hand undergoes initial training [100]. Other work has employed connectivity analyses, such as using Dynamic Causal Modeling (DCM) to probe specific interactions like those between the cerebellum and premotor cortex during adaptation [112]. While valuable, these approaches often focus on specific regions or pre-defined circuits. Yet, contemporary perspectives [1,113] suggest that it is functional interactions across distributed, large-scale networks that underpin learning-related processes. This shift in emphasis has sparked growing interest in analysis methods, such as the manifold learning approaches used here, which are capable of capturing large-scale changes in covariance patterns that may not be evident at the level of individual neurons or brain areas

[11,14,19,22,24]. Our current findings extend this body of work and suggest that many of the previously documented activation changes observed in individual brain regions—spanning cortical, subcortical, and cerebellar areas (see [7] for review)—may reflect subcomponents of a more extensive functional reorganization occurring across hierarchical brain systems. This perspective offers a more integrative, whole-brain view of how the central nervous system supports motor learning and generalization, emphasizing the importance of system-wide changes in FC rather than localized activity fluctuations [113,114].

A major finding from our study was that, whereas effector-related information was mainly encoded by selective changes in the manifold structure of lower-level contralateral sensorimotor regions, learning-related (effector-independent) information was mainly encoded by selective changes in higher-order transmodal cortex, particularly in regions of the DMN. This latter result is noteworthy for two reasons. First, it diverges from the emphasis of prior neurophysiological and fMRI studies, which have typically focused on the role of sensorimotor cortex in the learning process [7,14,16,111]. Second, it directly implicates the DMN—a collection of brain areas more commonly associated with abstract, self-referential mental processes [40,42,43,45,48]—in fundamental motor processes like adaptation and generalization. As we noted earlier, this finding aligns with the idea that explicit conscious processes, such as strategy use, underpin both higher-order cognitive functions [43,45,48] and motor learning processes [30,46,79]. Moreover, several distinct neuroanatomical and neurophysiological characteristics of DMN areas may allow it to play a unique role in supporting explicit strategies during learning.

First, the DMN occupies a unique topographic positioning on the cortical mantle, with each of its core constituent brain regions being located maximally distant from primary sensory and motor systems [13,48]. This neuroanatomical organization is thought to provide DMN areas broad oversight over whole-brain patterns of activity and allow it to engage in higher-order neural processes (e.g., the formation and implementation of strategies) that are independent of in-the-moment sensory inputs [48–116]. Second, as noted in our results, DMN areas exhibit lower myelin content [17,52,53], but more complex dendritic morphology [90–92,117,118]. As myelination can impede synapse formation [93], this dendritic complexity is thought to enhance the DMN's capacity for both learning and plasticity [93,94]. In addition, the DMN exhibits a higher density of neurotransmitter receptors per neuron than other cortical areas [52,53], a characteristic that confers upon the DMN a greater capacity for computational flexibility and cortical integration [52,53]. Finally, unlike sensorimotor areas that integrate neural information on a millisecond-to-second timescale, DMN areas appear to integrate information over much longer periods, ranging from seconds to minutes and beyond [17,49–51]. These extended temporal processing windows may enable DMN areas to maintain and implement explicit strategies that require sustained cognitive effort over multiple trials.

This leads to the important question: In what specific ways might the DMN support motor generalization? Based on previous functions ascribed to the DMN [40–42] and its unique neuroanatomical properties described above, one compelling idea is that the DMN performs an integrative function where it is critical for both evaluating the task context and subsequently reinstating the appropriate learned strategy.

First, successful generalization requires recognizing that the current situation—performing the task with the untrained hand—still shares the critical feature (presence of the VMR) that necessitates the learned adaptation strategy (i.e., explicit re-aiming). This process of context evaluation is essential; the brain must infer that the previously learned solution remains relevant despite the change in motor effector. Recent computational models, such as the COIN model [37], offer a compelling conceptual framework for how the brain might manage multiple motor memories, indicating that generalization may occur when the brain recognizes a similarity between the current context and the original learning setting. The DMN, with its well-documented roles in memory retrieval, contextual processing, and integrating internal states with external cues [40,43,45,44], seems ideally suited to perform this contextual inference function; i.e., assessing whether the current context warrants deploying the strategy learned during the initial adaptation phase.

Next, once the context is identified as appropriate, the learned strategy must be retrieved from memory and re-expressed. With respect to the implementation of strategies, prior neurophysiological studies in macaques have implicated

DMN neurons in the execution of response strategies during task-based learning [46,119]. This finding accords with the established role of several DMN areas in maintaining high-level, task-based representations for the purposes of planning and problem-solving [120–126]. With respect to memory retrieval, several studies have also implicated several areas of the DMN in this process, suggesting that retrieval involves the reinstatement of neural activity patterns observed during initial encoding [83,85,127–130]. Consistent with this work, we similarly found that the specific pattern of DMN manifold structure observed during early learning was selectively re-expressed during the early transfer phase (Fig 6). In summary, we posit that the DMN may facilitate both the recognition of the previously encountered context and the retrieval of learned strategies, thereby enabling individuals to generalize their motor behaviors to new contexts or effectors.

Given the role of strategy use in motor learning and generalization [28–30,33,35,79], another pertinent question is why other task-oriented networks, such as the Action Mode Network [131] or the Control network [70], were not as prominently featured in our analyses as the DMN. One possibility is that this apparent discrepancy reflects the specific nature of our manifold approach, designed to capture relatively large reconfigurations in whole-brain connectivity architecture across learning phases. It is conceivable that brain regions located in task-positive networks, while clearly essential for the moment-to-moment execution of motor actions and online error correction [131–134], might maintain a consistently high baseline of engagement and connectivity across the baseline, learning, and transfer phases of the task. Consequently, any changes in their connectivity patterns, as captured by manifold reorganization, might be less prominent compared to those observed in the DMN, where we found a notable re-expression of manifold structure during transfer. It is important to note, however, that task-positive networks were not entirely absent from our analyses; for instance, the Control-B network did rank as the third most modulated for the Task Epoch effect (Fig 5B), indicating some degree of involvement. Future investigations, perhaps employing methods sensitive to finer-grained temporal dynamics or regional activation patterns, could further elucidate the distinct contributions and interplay of both DMN and task-positive networks in supporting the multifaceted demands of motor learning and generalization.

## Methodological considerations

Whilst our study has identified common neural changes associated with initial adaptation and transfer, several methodological considerations warrant discussion. Firstly, our study exclusively examined the intermanual transfer of sensorimotor adaptation from the dominant to the non-dominant hand. It thus remains unclear if similar effects would emerge with the reversed training order. Our decision to study intermanual transfer in this single direction was informed by substantial evidence indicating that the transfer of learning from the non-dominant to dominant hand is highly variable across studies [56,103,135], with some cases showing no transfer at all [136]. Given this variability and the need for a reliable fMRI paradigm eliciting motor generalization, our experimental design represents a pragmatic compromise.

Secondly, because we assessed subjects' explicit strategy use at the end of the initial learning phase but before the transfer phase, it is possible that this arrangement may have biased subjects' behavior during transfer trials toward using an explicit strategy [33,35,137]. We placed report trials at this junction to (1) directly index subjects' knowledge about the VMR prior to the transfer phase, thus providing insight into the drivers of transfer performance, and (2) position them relatively equidistant from the learning and transfer phases. Nevertheless, we believe the influence of report trials on performance was relatively minimal, evidenced by (1) a strong positive correlation between early learning and early transfer performance, and (2) correlations between subjects' explicit reports and both early learning and transfer performance.

Finally, while manifold eccentricity offers a valuable and statistically tractable approach for characterizing the functional embedding of brain regions within a low-dimensional manifold space, it is important to acknowledge its inherent limitations as a single, summary measure. By design, eccentricity reduces the complex, multivariate patterns of regional connectivity to a single scalar value, inevitably averaging across a diverse array of underlying connectivity effects. This simplification, while facilitating group-level comparisons, necessarily entails a loss of information regarding the full richness and heterogeneity of dynamic network interactions. To mitigate this inherent limitation, our study also employed a complementary seed-based connectivity analysis to unpack the more granular patterns of inter-regional covariance changes that

contribute to the observed manifold eccentricity effects. Thus, while manifold eccentricity provides a powerful lens for examining macroscale shifts in functional segregation and integration, its interpretation should be considered in the context of these complementary, more fine-grained analyses.

## Conclusions

In the current study, we leveraged recent advances in dimension reduction techniques to investigate how brain regions reorganize their collective activity during sensorimotor adaptation and generalization. By projecting subjects' cortical, subcortical, and cerebellar FC patterns into a compact low-dimensional manifold space, we disentangled neural changes related to the effector being used from changes related to learning and generalization. Our main finding was that the transfer phase of sensorimotor adaptation reconstitutes the same patterns of manifold expansion and contraction in DMN areas that were observed during the initial adaptation phase. Together, these findings provide a novel characterization of the whole-brain macroscale changes associated with sensorimotor adaptation and generalization, and offer a unique perspective on the role of transmodal cortex, and the DMN in particular, in the transfer of motor learning.

## Materials and methods

### Participants

Forty-six right-handed individuals (27 females, aged 18−28 years) participated in the MRI study. Of these 46 participants, eight individuals were removed from the final analysis. Four of these participants were removed due to poor task compliance (i.e., > 25% of trials not being completed within the maximal trial duration) and/or missing data (i.e., > 20% of trials being missed due to insufficient pressure of the fingertip on the MRI-compatible tablet), and the remaining four subjects were removed due to excessive head motion in the MRI scanner and/or incomplete scans. We assessed right-handedness using the Edinburgh handedness questionnaire [138] and obtained written informed consent before beginning the experimental protocol. The Queen's University Research Ethics Board approved the study (reference #: CNS-019-16) and it was conducted in coherence to the principles outlined in the Canadian Tri-Council Policy Statement on Ethical Conduct for Research Involving Humans and the principles of the Declaration of Helsinki (1964).

### Procedure

Before undergoing MRI testing, participants first engaged in a training session inside a mock (0 T) scanner designed to replicate the appearance and sound of a real MRI scanner. This training session served several important purposes. Firstly, it familiarized participants with key features of the VMR task that they would later perform during the actual MRI scan. Secondly, it allowed us to screen participants for their ability to achieve baseline performance levels on the task. Thirdly, it assessed participants' ability to remain still for extended periods without experiencing claustrophobia. To monitor head movement during the training session, we attached a Polhemus sensor (Polhemus, Colchester, Vermont) to each participant's forehead using medical tape. This sensor provided real-time readouts of head displacement across three axes of translation and rotation (six dimensions total). Participants practiced task trials and underwent simulated anatomical scans while their head movements were tracked. If head translation or rotation exceeded 0.5 mm or 0.5° (within a pre-specified velocity criterion), an unpleasant auditory tone was delivered through a speaker system located near their head. Most participants learned to minimize head movement using this auditory feedback. Participants who met our criteria for task performance and head stability were subsequently invited to participate in the main study (details provided below). See our previous work for another description of these methods [24].

### Apparatus

During the mock (0 T) scanner testing, subjects performed target-directed hand movements by applying fingertip pressure on a digitizing touchscreen tablet (Wacom Intuos Pro M). For the actual MRI testing session, an MRI-compatible digitizing

tablet (Hybridmojo LLC, CA, USA) was used. In both the mock and real MRI scanners, visual stimuli were rear-projected using an LCD projector (NEC LT265 DLP, 1,024 × 768 resolution, 60 Hz refresh rate) onto a screen mounted behind the participant. Participants viewed the stimuli on the screen through a mirror attached to the MRI coil directly above their eyes, preventing them from seeing their hands. See our previous work for another description of these methods [24].

## Visuomotor rotation (VMR) task

To study sensorimotor adaptation and generalization, we employed the well-characterized VMR paradigm [1,54]. During the VMR task, participants performed blocks of trials using their left and their right index finger to execute center-out, target-directed movements. Following a series of baseline trials with each hand, we introduced a 45° clockwise rotation to the viewed cursor to investigate learning with the right hand. We selected this rotation magnitude specifically because it is known to robustly engage explicit adaptation strategies [28,33,35], which are the primary drivers of intermanual transfer in this paradigm [55,79], thus allowing us to effectively investigate the neural correlates of generalization. After this, we briefly assessed participants' re-aiming strategy associated with their learning. Finally, to study intermanual transfer, participants performed the same VMR task with their untrained left hand.

Each trial began with the participant moving a cursor (3 mm radius cyan circle) into a start position (4 mm radius white circle) at the center of the screen by sliding their index finger on a tablet. A ring centered around the start position indicated the distance between the cursor and the start position. The cursor became visible when its center was within 8 mm of the start position. After holding the cursor in the start position for 0.5 seconds, a target (5 mm radius red circle) appeared on a gray ring with a radius of 60 mm (the target distance) centered around the start position. The target appeared at one of eight randomized locations, separated by 45° increments (0, 45, 90, 135, 180, 225, 270, and 315°), in bins of eight trials. Note that the targets were presented sequentially, one per trial, and their order pseudorandomized within these 8-trial bins; this prevented participants from using a simultaneously visible adjacent target as a direct aiming guide and ensured that any strategy involving aiming towards a different location required a cognitive transformation of the current target goal. Participants were instructed to hit the target with the cursor by making a fast finger movement on the tablet, 'slicing' the cursor through the target to minimize online corrections during the reach. If the movement began before the target appeared (i.e., the cursor moved out of the start circle), the trial was aborted, and a "Too early" message appeared on the screen. In correctly timed trials, the cursor remained visible during the movement to the ring and then became stationary for one second, providing visual feedback of the endpoint reach error. If any part of the stationary cursor overlapped with the target, the target turned green to indicate a hit. Each trial ended after 4.5 s, regardless of whether the cursor reached the target. After a 1.5-s delay to save data, the next trial began with the presentation of the start position. See our previous work for similar methods [22,24,32].

During the participant training session in the mock MRI scanner (approximately two weeks before the VMR MRI testing session), participants performed a practice block of 80 trials (40 trials with each hand) with veridical feedback (no cursor rotation). This session familiarized participants with key task features (e.g., using the touchscreen tablet, trial timing, and using cursor feedback to correct errors) and allowed us to establish adequate performance levels. See our previous work for another description of this approach [24].

At the beginning of the MRI testing session, before the first scan, participants reacquainted themselves with the VMR task by performing 80 practice trials with veridical cursor feedback (40 trials with each hand). Next, we collected an anatomical scan, followed by four consecutive fMRI experimental runs. The first run included baseline blocks of 64 trials with veridical cursor feedback performed with the left hand. The second run was identical but performed with the right hand. The third run included a rotation block of 160 trials, where the cursor feedback was rotated clockwise by 45° during the reach, performed with the right hand. Prior to the fourth fMRI run, participants reported their strategic aiming direction over 16 trials. During these trials, a line between the start and target positions appeared and, using a separate MRI joystick (Current Designs) positioned at their left hip, participants rotated the line (using their left hand) to the direction they would

aim their finger movement to hit the target and clicked a button to confirm. The trial then proceeded as a normal reach trial using their right hand. Due to initial confusion among participants about the report trials, often failing to adjust the line during the first few trials, we discarded the first 8 report trials and calculated the mean aim direction using the final 8 trials (however, note that qualitatively identical results were obtained if all 16 trials were used). Finally, we resumed fMRI testing with the fourth run, which included a rotation block of 80 trials identical to the third run but performed with the left hand to investigate intermanual transfer. Note that participants were not informed about the nature or presence of the VMR before or during the experiment.

## MRI acquisition

Participants were scanned using a 3-Tesla Siemens TIM MAGNETOM Trio MRI scanner located at the Centre for Neuroscience Studies, Queen's University (Kingston, Ontario, Canada). All imaging was performed using a 32-channel head coil. At the beginning of the MRI testing session, we gathered high-resolution whole-brain T1-weighted (T1w) and T2-weighted (T2w) anatomical images (in-plane resolution 0.7 x 0.7 mm2; 320 x 320 matrix; slice thickness: 0.7 mm; 256 anterior and posterior commissure (AC-PC) transverse slices; anterior-to-posterior encoding; 2 × acceleration factor; T1w TR 2,400 ms; TE 2.13 ms; flip angle 8°; echo spacing 6.5 ms; T2w TR 3,200 ms; TE 567 ms; variable flip angle; echo spacing 3.74 ms).

Following this, we acquired 204, 204, 492, and 252 imaging volumes for the two baseline scans, and the learning and transfer functional scans, respectively. These functional MRI volumes were obtained using a T2*-weighted single-shot gradient-echo echo-planar imaging sequence (TR = 2,000 ms, slice thickness = 4 mm, in-plane resolution = 3 mm × 3 mm, TE = 30 ms, field of view = 240 mm × 240 mm, matrix size = 80 × 80, flip angle = 90°, and acceleration factor (iPAT) = 2 with GRAPPA reconstruction). Each volume comprised 34 contiguous (no gap) oblique slices acquired at a ~ 30° caudal tilt relative to the AC-PC plane, providing whole-brain coverage of the cerebrum and cerebellum. Additionally, each task-related scan included six extra imaging volumes at both the beginning and end of the scans for stabilization. See [24] for previous descriptions of these imaging methods.

## fMRI preprocessing

Preprocessing of anatomical and functional MRI data was performed using fMRIPrep 20.1.1 [139,140] (RRID:SCR_016216) which is based on Nipype 1.5.0 [141,142] (RRID:SCR_002502). Many internal operations of fMRIPrep use Nilearn 0.6.2 [143] (RRID:SCR_001362), mostly within the functional processing workflow. For more details of the pipeline, see the section corresponding to workflows in fMRIPrep's documentation. Below we provide a condensed description of the preprocessing steps (See [22,24] for previous descriptions of this same imaging approach).

T1w images were corrected for intensity non-uniformity (INU) with N4BiasFieldCorrection [144], distributed with ANTs 2.2.0 [145] (RRID:SCR_004757). The T1w-reference was then skull-stripped with a Nipype implementation of the antsBrainExtraction.sh workflow (from ANTs), using OASIS30ANTs as target template. Brain tissue segmentation of cerebrospinal fluid (CSF), white matter (WM), and gray matter (GM) was performed on the brain-extracted T1w using fast (FSL 5.0.9, RRID:SCR_002823) [146]. A T1w-reference map was computed after registration of the T1w images (after INU-correction) using mri_robust_template (FreeSurfer 6.0.1) [147]. Brain surfaces were reconstructed using recon-all (FreeSurfer 6.0.1, RRID:SCR_001847) [148], and the brain mask estimated previously was refined with a custom variation of the method to reconcile ANTs-derived and FreeSurfer-derived segmentations of the cortical GM of Mindboggle (RRID:SCR_002438) [149]. Volume-based spatial normalization to standard space (MNI152Nlin6Asym) was performed through nonlinear registration with antsRegistration (ANTs 2.2.0), using brain-extracted versions of both T1w reference and the T1w template.

For each BOLD run, the following preprocessing was performed. First, a reference volume and its skull-stripped version were generated using a custom methodology of fMRIPrep. The BOLD reference was then co-registered to the T1w reference using bbregister (FreeSurfer) which implements boundary-based registration [150]. Co-registration was configured with nine degrees of freedom to account for distortions remaining in the BOLD reference. Head-motion parameters with respect to the BOLD reference (transformation matrices, and six corresponding rotation and translation parameters) are estimated before any spatiotemporal filtering using mcflirt (FSL 5.0.9, [151]). BOLD runs were slice-time corrected using 3dTshift from AFNI 20160207 ([152], RRID:SCR_005927). The BOLD time-series were resampled to surfaces on the following spaces: fsaverage. The BOLD time-series (including slice-timing correction when applied) were resampled onto their original, native space by applying a single, composite transform to correct for head-motion and susceptibility distortions. These resampled BOLD time-series will be referred to as preprocessed BOLD in original space, or just preprocessed BOLD. The BOLD time-series were resampled into several standard spaces, correspondingly generating the following spatially-normalized, preprocessed BOLD runs: MNI152Nlin6Asym, MNI152Nlin2009cAsym. First, a reference volume and its skull-stripped version were generated using a custom methodology of fMRIPrep. Automatic removal of motion artifacts using independent component analysis (ICA-AROMA, [153]) was performed on the preprocessed BOLD on MNI space time-series after removal of non-steady state volumes and spatial smoothing with an isotropic, Gaussian kernel of 6 mm FWHM (full-width half-maximum). Corresponding "non-aggressively" denoised runs were produced after such smoothing. Additionally, the "aggressive" noise-regressors were collected and placed in the corresponding confounds file. Several confounding time-series were calculated based on the preprocessed BOLD: framewise displacement (FD), DVARS, and three region-wise global signals. FD and DVARS are calculated for each functional run, both using their implementations in Nipype (following the definitions by [154]). The three global signals are extracted within the CSF, the WM, and the whole-brain masks. Additionally, a set of physiological regressors were extracted to allow for component-based noise correction (CompCor, [155]). PCs are estimated after high-pass filtering the preprocessed BOLD time-series (using a discrete cosine filter with 128s cut-off) for the two CompCor variants: temporal (tCompCor) and anatomical (aCompCor). tCompCor components are then calculated from the top 5% variable voxels within a mask covering the subcortical regions. This subcortical mask is obtained by heavily eroding the brain mask, which ensures it does not include cortical GM regions. For aCompCor, components are calculated within the intersection of the aforementioned mask and the union of CSF and WM masks calculated in T1w space, after their projection to the native space of each functional run (using the inverse BOLD-to-T1w transformation). Components are also calculated separately within the WM and CSF masks. For each CompCor decomposition, the $k$ components with the largest singular values are retained, such that the retained components' time series are sufficient to explain 50 percent of variance across the nuisance mask (CSF, WM, combined, or temporal). The remaining components are dropped from consideration. The head-motion estimates calculated in the correction step were also placed within the corresponding confounds file. The confound time series derived from head motion estimates and global signals were expanded with the inclusion of temporal derivatives and quadratic terms for each [156]. Frames that exceeded a threshold of 0.5 mm FD or 1.5 standardized DVARS were annotated as motion outliers. All resamplings can be performed with a single interpolation step by composing all the pertinent transformations (i.e., head-motion transform matrices, susceptibility distortion correction when available, and co-registrations to anatomical and output spaces). Gridded (volumetric) resamplings were performed using antsApplyTransforms (ANTs), configured with Lanczos interpolation to minimize the smoothing effects of other kernels [157]. Non-gridded (surface) resamplings were performed using mri_vol2surf (FreeSurfer).

**Regional time series extraction.** For each participant and scan, the average BOLD time series were computed from the grayordinate time series for (1) each of the 400 regions defined according to the Schaefer 400 parcellation [58]; (2) each of the 32 subcortical regions defined according to the Tian scale II, 3T subcortical atlas [59], which included anterior hippocampus, posterior hippocampus, dorsoanterior thalamus, ventroanterior thalamus, dorsoposterior thalamus, ventroposterior thalamus, lateral amygdala, medial amygdala, anterior caudate, posterior caudate, nucleus accumbens-shell, nucleus accumbens-core, anterior

putamen, posterior putamen, anterior globus pallidus, posterior globus pallidus; and (3) each of the 32 regions defined according to Nettekoven cerebellar atlas [60] with medium granularity which included four motor regions (M1, M2, M3, and M4), four action regions (A1, A2, A3, and A4), four demand regions (D1, D2, D3, and D4), and four sociolinguistic regions (S1, S2, S3, and S4). Region time series were denoised using the above-mentioned confound regressors in conjunction with the discrete cosine regressors (128s cut-off for high-pass filtering) produced from fMRIprep and low-pass filtering using a Butterworth filter (100s cut-off) implemented in Nilearn. Finally, all region time series were z-scored. See our previous work [22,24] for similar approaches.

### Neuroimaging data analyses

**Covariance estimation and centering.** For every participant, region time series from the functional scans were spliced into six equal-length task epochs (96 imaging volumes each), after having discarded the first 6 imaging volumes (thus avoiding scanner equilibrium effects). This allowed us to estimate patterns of FC from continuous brain activity over the corresponding 32 trials (four 8-trial bins) for each epoch; Left and Right Baseline each comprised of the initial 32 trials; and each of Right Early and Late Learning and Left Early and Late Transfer, each comprised of the initial and final 32 trials associated with phase of the task, respectively (see Fig 1B). Then, we separately estimated FC matrices for each epoch by computing the region-wise covariance matrices using the Ledoit–Wolf estimator [158]. Note that our use of equal-length epochs for the six phases ensured that no biases in covariance estimation were introduced due to differences in time series length.

Next, we centered the connectivity matrices using the approach advocated by [64], which leverages the natural geometry of the space of covariance matrices [32,63]. In brief, this involved adjusting the covariance matrices of each participant to have a common mean, which was equivalent to the overall mean covariance, thus removing subject-specific variations in FC. First, a grand mean covariance matrix, $\overline{S}_{gm}$,, was computed by taking the geometric mean covariance matrix across all participants and epochs. Then, for each participant $i$, we computed the geometric mean covariance matrix across task epochs, $\overline{S}_i$, and each task epoch covariance matrix $S_{ij}$ was projected onto the tangent space at this mean participant covariance matrix $\overline{S}_i$ to obtain a tangent vector

$$T_{ij} = \overline{S}_i^{\frac{1}{2}} \log \left( \overline{S}_i^{-\frac{1}{2}} S_{ij} \overline{S}_i^{-\frac{1}{2}} \right) \overline{S}_i^{\frac{1}{2}},$$

where $log$ denotes the matrix logarithm. We then transported each tangent vector to the grand mean $\overline{S}_{gm}$ using the transport proposed by [64], obtaining a centered tangent vector

$$T_{ij}^c = GT_{ij}G^T,$$

where $G = \overline{S}_{gm}^{1/2}\overline{S}_i^{-1/2}$. Finally, we projected each centered tangent vector back onto the space of covariance matrices, to obtain the centered covariance matrix

$$S_{ij}^c = \overline{S}_{gm}^{\frac{1}{2}} \exp \left( \overline{S}_{gm}^{-\frac{1}{2}} T_{ij}^c \overline{S}_{gm}^{-\frac{1}{2}} \right) \overline{S}_{gm}^{\frac{1}{2}},$$

where $exp$ denotes the matrix exponential. For the benefits (and general necessity) of this centering approach, see Fig 1, and for an additional overview, see [32]. See also our previous work [22,24] for similar approaches.

**Manifold construction.** Following the estimation and centering of epoch-specific covariance matrices, our subsequent analyses utilized manifold learning techniques to characterize large-scale network reorganization [13,18,17,12]. This analysis pipeline, focusing on the whole-brain FC structure captured in covariance matrices derived from task-based BOLD signals [22,24–26,32,102,159–162], was specifically chosen for its ability to provide a data-driven, whole-brain perspective on distributed network dynamics underlying learning and generalization. By embedding the connectivity profile of all brain regions into a low-dimensional space, this approach allows for the investigation of emergent macroscale shifts in functional architecture without the prerequisite of defining specific seed regions or pre-specifying network models,

offering insights complementary to methods primarily focused on univariate activation or specific task-modulated pairwise interactions (e.g., Psychophysiological Interaction, DCM [163,164]). The procedures for manifold construction based on these centered covariance matrices are detailed below.

Connectivity manifolds for all centered FC matrices were derived through the following steps. First, consistent with previous studies [13,22,67,12], we applied row-wise thresholding to retain the top 10% of connections in each row. We then computed the cosine similarity between each row to produce an affinity matrix, which describes the similarity of each region's connectivity profiles. Next, we performed PCA to obtain a set of PCs that provide a low-dimensional representation of the connectivity structure (i.e., connectivity gradients). We selected PCA as our dimensionality reduction technique based on recent research indicating that PCA offers improved reliability over non-linear dimensionality reduction methods [67].

To provide a basis for comparing changes in functional network architecture that arise during learning, specifically, we constructed a common template manifold, using the same aforementioned manifold construction procedures, from a group-averaged left- and right-hand baseline connectivity matrix. This common baseline connectivity matrix was derived by first computing, within each subject, the mean of the Left and Right Baseline centered connectivity matrices, and then subsequently computing the geometric mean across these averaged Baseline matrices. We aligned all individual manifolds (38 participants×6 epochs; 228 total) to this common baseline template manifold using Procrustes alignment. All analyses on the aligned manifolds were performed using the top three PCs, which cumulatively explained 50.90% of the total variance in the template manifold (however, note that using a larger number of components had negligible effects on the overall results, see Fig A in S1 Text). Together, this approach enabled us to uniquely examine both effector-specific and effector-independent, learning-related changes in manifold structure (across all six epochs), and specifically how these deviate from the Baseline manifold functional architecture. See our previous work [22,24] for a description of similar approaches.

**Manifold eccentricity.** Recent work has quantified the embedding of brain regions in low-dimensional connectivity spaces using Euclidean distance [22–24,68,69,165]. Here, we calculated "Manifold Eccentricity" as the Euclidean distance between a single region's coordinates in the aligned 3D manifold space and the manifold centroid (origin: [0,0,0]) [23]. This provides a scalar index quantifying how typical or atypical a region's whole-brain FC profile is relative to the average profile represented by the centroid. Regions with highly distinct connectivity patterns (often reflecting functional specialization) tend to lie further from the centroid, resulting in higher eccentricity, while regions sharing connectivity features across multiple systems lie closer, yielding lower eccentricity. Thus, in network neuroscience terms, higher eccentricity generally indicates greater functional segregation (a more distinct profile, often with strong within-system coupling), while lower eccentricity suggests greater functional integration (a profile more similar to the average, reflecting broader cross-system coupling) [23,68,69]. Importantly, eccentricity reflects the relative distinctiveness of a connectivity pattern within the manifold, not simply overall connectivity strength or magnitude. To empirically validate this interpretation within our data, we confirmed that baseline eccentricity significantly correlated with established graph theoretical measures of network topology (see Fig B in S1 Text), supporting its use as an index of functional segregation versus integration. These measures were calculated on the row-wise thresholded template connectivity matrix and included node strength, which is the sum of a region's connectivity weights; within-module degree z-score, which measures the degree centrality of a region within its respective network; and participation coefficient, which measures the network diversity of a region's connectivity distribution [166]. Cortical regions were assigned to their respective intrinsic functional networks [58,70] for these different graph theoretic calculations. For our main analyses, we computed the manifold eccentricity for each brain region, participant, and epoch. This allowed us to statistically test for manifold expansions (increases in eccentricity) and contractions (decreases in eccentricity) throughout early and late learning and transfer trials. See our previous work [22,24] for a description of similar approaches.

**Representational similarity analysis.** Using established methods [74], we analyzed six epochs as conditions and the eccentricity values for 464 regions spanning the cortex, subcortex, and cerebellum as data channels to compute a RSM. This RSM measured pairwise similarity between conditions for each subject using Pearson's r correlation on the whole-brain eccentricity data (see Fig 3). Subsequently, we defined four RSM models representing different possible main effects in the data: Epoch (Baseline versus Early versus Late), Hand (Left versus Right), Learning (Baseline versus Learning conditions) and Time (where time-adjacent scans were coded as being similar). To test the explanatory power of each model, we performed a second-level analysis by comparing the calculated RSMs for each subject against the models' RSMs using cosine similarity. We computed the uncertainty of model-performance estimates by bootstrapping subjects, thereby gauging the variance expected from repeating the experiment with new subjects. [Note that since all conditions in our experiment were considered, bootstrapping conditions were not unnecessary]. Follow-up two-tailed *t* tests were conducted to compare the performance of the models to each other, with corrections for multiple comparisons applied using the FDR correction method ($q < 0.05$). Additionally, we performed one-sided comparisons of each model's performance against zero and against the lower-bound estimate of the noise ceiling (an estimate of the performance the true model would achieve), applying Bonferroni correction for multiple comparisons.

**Eccentricity analyses.** To compare the eccentricity of regions across different task epochs, we conducted a $3 \times 2$ rmANOVAs for each region, with Task Epoch (Baseline, Early, and Late) and Hand (Left versus Right) as within-subject effects. Results were corrected for multiple comparisons using an FDR correction ($q < 0.05$) across all comparisons (464 regions × 3 contrasts for epoch, hand, and the interactions of effects). We performed follow-up *post hoc* paired-samples *t* tests on significant regions to identify significant changes in eccentricity between individual epochs/conditions. See our previous work [22,24] for a description of similar approaches.

**Seed connectivity analyses.** In order to understand the underlying changes in regional covariance that ultimately give rise to the observed changes in manifold eccentricity, we performed a series of seed connectivity contrasts. To this end, we selected several seed regions that showed statistically significant main effects in our rmANOVAs (i.e., main effects of Task Epoch and Hand). For each selected seed region exemplifying a main effect of Task epoch, we generated, for each subject, averaged seed-based connectivity maps for each of the Baseline (average of LH and RH Baseline), Early (average of Early RH Learning and LH Transfer), and Late (average of Late RH Learning and LH Transfer) epochs, and then performed paired-samples *t* tests across subjects between epochs. For each selected seed region exemplifying a main effect of Hand, we generated, for each subject, averaged seed-based connectivity maps for each of the RH (Baseline, Early, and Late learning) and LH (Baseline, Early, and Late transfer) conditions, and then performed a paired-samples *t* test between those conditions. For all contrasts, we opted to show the unthresholded t-maps to visualize the complete multivariate pattern of connectivity changes that drive changes in manifold eccentricity (a multivariate measure). In addition, for each region, we used scatterplots to visualize the pattern of eccentricity change throughout all tasks. Where relevant, we also constructed spider plots for each seed region to further characterize these changes at the network-level, by averaging the t-values across individual regions according to their network assignment [70]. Note that these analyses are mainly intended to provide visual characterizations (and interpretations) of the connectivity changes of representative regions exhibiting the main effects. See our previous work [22,24] for a description of similar approaches.

**Neurosynth meta-analyses.** To explore the most prominent behavioral-cognitive processes associated with our two main-effects F-stat brain maps (for Hand and Task Epoch), we utilized the correlation decoder tool in the NiMARE library [75]. NiMARE is a comprehensive Python package designed for conducting meta-analyses and derivative analyses of neuroimaging data. Unlike other meta-analytic packages with limited algorithms, NiMARE offers a standardized syntax for a broad range of analyses and integrates seamlessly with prominent databases of coordinates and images from fMRI studies. For our analyses, we selected the Neurosynth [76] database and performed our meta-analysis using the correlation decoder tool in the MNI-152 space. This tool generates a set of terms associated with a spatial map. The results of this analysis were visualized as word clouds (see Fig 4E and 4F) using custom Python scripts. To refine the

results, we excluded terms referring to neuroanatomy (e.g., "inferior" or "sulcus") and removed repeated terms (e.g., "semantic" and "semantics").

**Examination of structure-function relationships.** To investigate the relationship between our main effect F-stat brain maps and different properties of cortical structure, we utilized two existing brain maps: the Human Connectome Project 1200 (HCP) myelin map (T1w/T2w ratio) from the Neuromaps toolbox [98] and the first principal gradient of receptor density from Hansen and colleagues [52]. Both maps were in the fsLR coordinate system with 32k vertices per hemisphere. For the receptor density map, this involved parcellating the PET maps of 19 neurotransmitter receptors and transporters from Hansen and colleagues [52] to the Schaefer 400 atlas in MNI-152 space. We then created a z-scored region-by-receptor matrix using weighted averages of maps with the same tracer, and then calculated the first PC via PCA, and projected it onto a 32k-vertex fsLR mesh [52]. For comparison, we projected the *F*-values from our two main effects brain maps (Fig 4A and 4B) from Schaefer 400 space onto the 32k-vertex fsLR mesh and then correlated them with the HCP-1200 myelin map and the calculated first PC of receptor density using Pearson's r correlation. To test for statistical significance, using established methods, we constructed spatial null distributions by performing 1,000 iterations of the Váša spin-testing permutation procedure [98,97]. We then empirically assessed the statistical significance of our true correlation values against this spatial null distribution for each F-stat map and each map of cortical structure (at $p < 0.05$, see Fig 6).

**General Linear Model (GLM) Analyses.** To assess task-related changes in regional BOLD activity complementary to our primary connectivity analysis, we applied a standard mass-univariate GLM. For this analysis, to ensure comparability, we used the same preprocessed, parcellated fMRI time series data spanning the cortex (Schaefer 400 cortical parcellation [58]), subcortex (Tian subcortical (scale II) 3T subcortical atlas [59]), and cerebellum (Nettekoven cerebellar atlas [60]). The design matrix for each participant included regressors for each of the six task epochs used in the connectivity analyses (LH Baseline, RH Baseline, RH Learning Early, RH Learning Late, LH Transfer Early, LH Transfer Late). Each epoch was modeled as a sustained block encompassing the relevant 32 trials (96 TRs) and convolved with the canonical Glover hemodynamic response function along with its first temporal derivative [167,168] to account for latency variations (12 regressors total). A constant term (intercept) representing the session mean was also included. Subject-level beta estimates for each condition were estimated using ordinary least squares regression, implemented via the nilearn *run_glm* function. Group-level statistical analyses were conducted using paired-samples *t* tests on the beta estimates for several predefined contrasts of interest relevant to learning and transfer (e.g., RH Learning Early > RH Baseline, LH Transfer Early > LH Baseline, LH Transfer Early > RH Learning Early, and LH Transfer Early > RH Learning Late), as detailed in Fig E in S1 Text. Resulting group-level statistical maps (*z*-scores) were corrected for multiple comparisons across all 464 brain regions using the Benjamini–Hochberg FDR procedure ($q < 0.05$). To assess task-related modulations in regional BOLD activity, we conducted a $3 \times 2$ rmANOVAs on the GLM beta estimates for each region, with Task Epoch (Baseline, Early, and Late) and Hand (Left versus Right) as within-subject factors. Multiple comparisons were corrected using FDR correction ($q < 0.05$) across all tests (464 regions $\times$ 3 effects: main effects of Epoch and Hand, and their interaction). For regions exhibiting significant main effects, we performed follow-up *post hoc* paired-samples *t* tests to compare Right versus Left hand (for Hand main effec*t*), and Early versus Baseline and Late versus Early epochs (for Epoch main effect), identifying specific condition-driven changes in BOLD activation.

## Behavioral data analysis

**Data preprocessing.** Trials in which the reach was initiated before the target appeared (~4% of trials) or in which the cursor did not reach the target within the time limit (~5% of trials) were excluded from the offline analysis of hand movements. As insufficient pressure on the touchpad resulted in a default state in which the cursor was reported as lying in the top left corner of the screen, we excluded trials in which the cursor jumped to this position before reaching the target

region (~2% of trials). We then applied a conservative threshold on the movement and reaction times, removing the top 0.05% across all subjects. As the task required the subject to determine the target location prior to responding, we also set a lower threshold of 100 ms on the reaction time (see our previous work [22,24] for a description of similar approaches). We then measured visuomotor error on each trial by computing the angular difference between the target location and the final cursor position. We also estimated subjects' re-aiming strategy from the final eight reporting trials by computing the mean report angle (relative to the target).

**Behavioral correlation analysis.** To investigate the relationship between changes in manifold structure and individual differences in task performance, we computed two correlations across participants. The first correlation was between each region's eccentricity change from RH Baseline to RH Learning Early and the median angular error of the first 32 learning trials. The second correlation was between each region's eccentricity change from LH Baseline to LH Transfer Early and the median angular error of the first 32 transfer trials (see Fig 9). The resulting correlation maps were FDR-corrected at $q < 0.05$. These effects were also evaluated at the network level by mapping each region to its corresponding functional network assignment for each hemisphere and computing the mean eccentricity change for each network, then correlating the change from Baseline to Early for each network with subjects' median angular errors. To determine if the observed correlations could be attributed to chance due to spatial autocorrelations in the brain maps, we performed 1,000 spin-tests [98,97] for each functional network. This allowed us to generate a topographical distribution of correlations expected purely by chance. We empirically assessed the statistical significance of our actual correlation values against this spatial null distribution for each brain network.

To explore the underlying changes in FC that give way to these network-behavior correlations, we performed seed-network connectivity contrasts between the Baseline and Early phases (for the relevant RH and LH conditions) and correlated these changes with subject's corresponding performance (i.e., median angular error during the relevant condition). As with the 'Seed Connectivity Analyses' section above, we have shown the unthresholded correlation maps to allow for visualization of the complete multivariate pattern of between-network correlations that underlies the manifold-level correlations (Fig 8). Together, these complementary approaches enabled us to explore how individual differences in performance relate to changes in manifold structure at both the region- and network-levels. See our previous work [22,24] for a description of similar approaches.

## Software

Imaging data were preprocessed using fmriPrep [139], which is open source and freely available. All analyses were performed using Python 3.11.6 and involved the following open-source Python packages. Functional connectivity estimation and centering were performed with Nilearn 0.10.2 [143] and PyRiemann 0.2.6 [169], respectively. All steps to generate and align connectivity manifolds were carried out using Brainspace 0.1.1 [12]. Graph theoretical measures were computed using the Brain Connectivity Toolbox (bctpy; https://github.com/aestrivex/bctpy/wiki), and spin permutation testing procedures were implemented in neuromaps 0.0.5 [98]. All statistical analyses were performed with Pingouin 0.5.4 [170] and SciPy 1.11.4 [171]. For unsupervised learning analyses, UMAP was implemented with Umap-learn 0.5.5 [71], and k-means clustering was performed with Scikit-learn 1.3.2 [172]. Surface visualizations were generated using Surfplot 0.2.0 [173]. Representational similarity analysis was conducted using rsatoolbox 0.1.5 [74], and meta-analysis was generated using the NiMare library 0.2.2 [75]. Task-related modulations in regional BOLD activity were assessed using a General Linear Model (GLM) implemented with the Nilearn 0.10.2 [143]. All other general data processing and visualization were performed using NumPy 1.26.2 [174], Pandas 2.2.1 [175], Nibabel 5.2.0 [176], Matplotlib 3.8.2 [177], Seaborn 0.13.2 [178], and Cmasher 1.6.3 [179]. Archived code for performing the analyses can be found at https://zenodo.org/records/15651833.

## Supporting information

**S1 Text.** Fig A in S1 Text: Comparison of effector- and learning-related changes in manifold eccentricity using different numbers of principal components (PCs). **Fig B:** Functional connectivity properties that underlie manifold eccentricity. **Fig C:** Brain regions showing a Task Epoch $\times$ Hand interaction effect. **Fig D:** Effector- and learning/transfer-related changes

in connectivity do not reflect univariate changes in brain activity. **Fig E:** Task-related modulations in regional BOLD activity revealed by GLM analysis. **Fig F:** Relationship between transfer and learning-related changes in eccentricity. **Fig G:** Relationship between learning performance and learning-related changes in eccentricity.
(DOCX)

## Acknowledgments

The authors would like to thank Martin York and Sean Hickman for technical assistance, and Don O'Brien for assistance with data collection.

## Author contributions

**Conceptualization:** Corson N. Areshenkoff, Anouk J. De Brouwer, J. Randall Flanagan, Jason P. Gallivan.

**Data curation:** Corson N. Areshenkoff, Jason P. Gallivan.

**Formal analysis:** Ali Rezei.

**Funding acquisition:** Jason P. Gallivan.

**Investigation:** Ali Rezei, Corson N. Areshenkoff, Anouk J. De Brouwer, J. Randall Flanagan, Jason P. Gallivan.

**Methodology:** Corson N. Areshenkoff, Anouk J. De Brouwer, Joseph Y. Nashed.

**Project administration:** Jason P. Gallivan.

**Resources:** Ali Rezei, Corson N. Areshenkoff, Daniel J. Gale, Anouk J. De Brouwer, Jason P. Gallivan.

**Software:** Ali Rezei, Daniel J. Gale.

**Supervision:** J. Randall Flanagan, Jason P. Gallivan.

**Validation:** Jason P. Gallivan.

**Visualization:** Ali Rezei.

**Writing – original draft:** Ali Rezei, Jason P. Gallivan.

**Writing – review & editing:** Ali Rezei, Corson N. Areshenkoff, J. Randall Flanagan, Jason P. Gallivan.

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
