## [Editor Report · Decision Letter 0]

Dear Jason, 

Thank you for submitting your manuscript entitled "Human cortical-subcortical manifold structure during the transfer of motor learning" for consideration as a Research Article by PLOS Biology.

Your manuscript has now been evaluated by the PLOS Biology editorial staff as well as by an academic editor with relevant expertise and I am writing to let you know that we would like to send your submission out for external peer review.

Once your full submission is complete, your paper will undergo a series of checks in preparation for peer review. After your manuscript has passed the checks it will be sent out for review. To provide the metadata for your submission, please Login to Editorial Manager (https://www.editorialmanager.com/pbiology) within two working days, i.e. by Dec 05 2024 11:59PM.

Kind regards,

Christian

Christian Schnell, PhD, 

Senior Editor

PLOS Biology

cschnell@plos.org

---

## [Decision Letter · Decision Letter 1]

Dear Jason,

Thank you for your patience while your manuscript "Human cortical-subcortical manifold structure during the transfer of motor learning" was peer-reviewed at PLOS Biology. It has now been evaluated by the PLOS Biology editors, an Academic Editor with relevant expertise, and by several independent reviewers. 

In light of the reviews, which you will find at the end of this email, we would like to invite you to revise the work to thoroughly address the reviewers' reports.

As you will see below, the reviewers were overall very interested in your study, but they all raised some methodological and interpretational concerns. R2 and R3 also mentioned that the clarity of the presentation needs to be improved and that some of the analyses are too abstract. Based on these reports, we encourage you to focus in your revision on providing additional support by including additional analyses (for example, more traditional fMRI measures such as GLM-based activation patterns and MVPA as requested by all reviewers) and on carefully interpreting your data. We do not think that additional experimental evidence is required, but if you have the data, we would of course be happy for you to include it. 

Given the extent of revision needed, we cannot make a decision about publication until we have seen the revised manuscript and your response to the reviewers' comments. Your revised manuscript is likely to be sent for further evaluation by all or a subset of the reviewers.

**IMPORTANT - SUBMITTING YOUR REVISION**

*Re-submission Checklist*

*Published Peer Review*

*PLOS Data Policy*

*Blot and Gel Data Policy*

Sincerely,

Christian

Christian Schnell, PhD,

Senior Editor

PLOS Biology

cschnell@plos.org

REVIEWS:

Reviewer #1: The authors investigated how inter-manual transfers of motor learning, utilizing fMRI and a visuomotor adaptation task. Key findings reveal that the brain's default mode network (DMN), a core of trans-modal areas, reactivates similar activity patterns during the transfer phase as observed during initial learning. These changes are linked to structural properties of the DMN, including myelin content and neurotransmitter receptor density. This research highlights the DMN's critical role in generalizing motor learning across untrained contexts, emphasizing the interplay of higher-order brain areas in sensorimotor learning and generalization.

This paper is distinguished by its well-thought-out experimental design and the use of advanced and diverse techniques for brain data analyses. It effectively captures whole-brain activity changes during motor learning.　Their complex analyses are described concisely and clearly, making them easy to understand.

One concern is whether the reactivation of the DMN is specific to intermanual transfer or whether it is a phenomenon commonly observed in the early stages of learning. To answer this question, the relationship between intermanual transfer and DMN reactivation would need to be investigated in more detail.

The authors investigated a correlation between the initial error in the transfer session and the eccentricity of the DMN (Figure 9). However, the initial error in the transfer session alone cannot serve as a direct indicator of intermanual transfer. Ideally, a pretest-posttest paradigm should be used, measuring the initial error with the left hand under visuomotor rotation and then assessing how much this error decreases after training with the right hand under visuomotor rotation.

As the experimental design does not follow this paradigm, an alternative analysis could be to calculate the intermanual transfer rate by comparing the errors between the hands. For example, the percentage increase in the left hand's initial error during the transfer epoch compared to the right hand's final error during the learning epoch could be examined. In this case, a smaller increase in error would indicate a higher degree of inter-manual transfer.

A minor point is that the cells in the representational similarity matrices in Figures 3C and 3D show values of 0 and 1, which was difficult to understand. If these represent similarity, they should be continuous values. It is unclear whether thresholding has been applied or whether the 0 and 1 values indicate whether statistical significance has been exceeded. No explanation was found in the main text or methods section.

Reviewer #2: [NOTE: I reviewed this paper for another journal. My review below echoes some of my previous thoughts, with updates made where appropriate.]

In this study, Rezaei et al. combine functional neuroimaging with a visuomotor adaptation inter-manual transfer task. The authors investigated low dimensional structure in brain-wide connectivity patterns during adaptation, specifically in the case of the transfer of motor learning across hands. The authors were interested in how changes in connectivity structure may be similar (or different) when a learned motor behavior is expressed with an effector that did not undergo initial learning (i.e., effector independence). The authors used cutting-edge fMRI data analysis techniques that allow them to compute patterns of "eccentricity" of particular networks in a low-d connectivity manifold. The authors observed that the structure of these connectivity manifolds was similar across the expression of motor learning for both hands, specifically in areas of association cortex and the default mode network (DMN). This neural connectivity similarity between initial learning and transfer phases was not seen in sensorimotor regions. This result builds upon previous work by this group (Gale et al., '22), linking transmodal and DMN areas to explicit motor learning strategies, which, crucially, are the primary (and perhaps only) driver of inter manual transfer in visuomotor adaptation. Follow-up analyses suggest that the observed neural dissociations may relate to biological/structural properties of cortex, and link DMN connectivity to between-subjects differences in learning and transfer.

Strengths:

-Overall this is a sophisticated fMRI methodological study and the behavior is nicely characterized. The analysis methods are quite impressive and appear to be rather robust; the neural results are thus generally rather clean and convincing, and multiple control analyses are implemented throughout to help buttress the main empirical claims.

-The addition of a transfer phase in the VMR task is a novel addition that builds upon previous similar work by the group (Gale et al 2022), extending the work to a more direct measure of explicit strategic learning.

-The consistent use of effector-specific (hand-based) analyses throughout the paper helps establish sanity checks and expected "ground truths" (i.e., lateralization effects), on which to support the key learning/transfer related results. This helps with the rigor of the study.

-The addition of subcortical analyses and findings is interesting and broadens the study's scope beyond oft-studied cortical networks/regions. 

-The figures and visualizations are generally very thorough, informative, and nicely designed.

Weaknesses:

- Interpretation/Impact: The two interpretations of the results offered on page 36 - strategy reinstatement and contextual inference - seem like two sides of the same coin (i.e., you have to infer a similar context during early/re-learning to retrieve the correct explicit strategy). Thus it is not totally clear what the key distinctions are w/r/t interpretations of the data (moreover, the COIN model is focused on implicit motor learning, whereas as here explicit learning is studied). This leads to a more general vagueness, which is that it is not clear how the main DMN connectivity analyses relate to the relevant concepts of strategy formation or memory reinstatement. For the latter at least, there is a large fMRI literature in the episodic memory domain that tackles exactly these questions - i.e., how similar is a multivariate neural pattern in context A (e.g., early learning) versus context B (e.g., late learning). Perhaps by shuffling different learning epochs and looking at multivariate pattern correlations, the authors could do MVPA analyses across DMN regions and actually test for reinstatement-like effects? This would be helpful as it would be a less abstracted analysis of cortical activity and help build more specific interpretations alongside the connectivity results.

-The analyses tend to be quite abstracted from more traditional fMRI measures. These complex, interesting metrics are a feature of the study to be sure, but it would be helpful to have more declarative sentences throughout to really guide the reader about how to think of each result in terms of what are fundamentally higher-order consequences of regional correlations in BOLD time series. These abstractions also made me wonder how much within-run changes in gross univariate BOLD signal could affect the manifold/eccentricity metrics via more low-level statistics, like signal increases or even changes in the SNR? For instance, early learning/transfer are cognitively more demanding than other epochs, which would have large univariate effects throughout the brain.

-It seems possible that other more task-positive networks could have shown up in these analyses, like the "Action Mode Network" (similar to the CON network, Dosenbach et al 2025). It would be interesting for the authors to discuss not just why the DMN showed up, by why more task-oriented networks did not, given the active, externally-focused nature of strategic motor learning.

Minor Things:

-There is a sentence repeated before the last paragraph on page 35

-Sometimes figure panels are cited in odd places relative to the flow of the text (e.g., fig 4D). These could be cited earlier or moved to a lower part of the figure?

-It is not exactly clear what the word-cloud analyses add? These could be motivated more.

Reviewer #3: This is fascinating, and methodologically very elaborate, human neuroimaging study of transfer of motor learning (from right to left hand) using sensorimotor adaptation task. The functional connectivity analysis across different epochs of the task suggests that the transfer of learning across the hands is supported by similar patterns of activity as those that underlie the initial learning, in the "default mode network" (DMN) regions. 

There is much to like about the study. The Introduction and Discussion are extremely clear and very much to the point. Generally, the paper is very well written. The methods are highly elaborate and well documented, and as much as I am able to judge various pipelines, very rigorous. The results, taken at face value, are very interesting. However, I found the presentation of results extremely dense and at times too abstract, making it very hard to follow and evaluate the main messages of the study. In addition, I have a number of methodological comments. 

Specifically, I am listing four main topics in terms of clarifications/questions, which are primarily related to the issues of understanding and possibly interpretation. But I think it would be important to clarify these concerns also for the benefit of a broad readership of the PLoS Biol. 

1) The nature of the task. Given what is probably highly noticeable dissociation between intended and observed movements (45 deg rotation), this is arguably a case of very explicit adaptation, at least in those subjects who showed explicit re-aiming in the report trials. Why so large rotation has been used?

A related concern with the task is that 45 deg rotation apparently matches 45 deg between 8 possible target positions around the circle, potentially eliciting a high level abstract strategy to aim for the "preceding counterclockwise" target. Was there any debriefing of participants, outside of report trials, and have authors considered this possibility?

I really like the analysis in Figure 8. At least in the subjects that showed some rotation awareness and explicit strategic adjustment, it makes sense that self-referential functions supported by DMN are engaged. But it was not readily clear how to relate the inter-subject differences in explicit learning to the underlying neural connectivity patterns in Figure 9 (please see below).

More generally, the task is quite simple and the level of "generalization" (i.e. aim with the fixed offset) is fairly trivial, especially for participants that show explicit understanding. Why learning and transfer of such a task should involve apparently profound "reorganization" of connectivity, and how persistent such reorganization might be? 

2) It is functional connectivity study, using, at the core, inter-regional correlations (or covariance) matrices of ROI-based BOLD time series. The methods explain in some detail a denoising pipeline using confound regressors using pre-packaged preprocessing pipeline. But what about the task-related regressors? The typical application of such functional connectivity approaches mainly focuses on resting-state data, where there is no task-specific variance. There are several methods for calculating functional or effective connectivity in task-related data, but to the best of my understanding, they all include, in one way or another, task regressors to account for activation-induced changes in connectivity. 

3) My main concern is that the current presentation of results is hard to follow, especially for a non-expert in advanced dimensionality reduction methods. In particular, the meaning of manifold eccentricity, on which many results are based, need to be clarified and explained in a more consistent manner.

To start, the sentence in the abstract, "Here we show, using human functional MRI and a sensorimotor adaptation task involving the transfer of learning from the trained to untrained hand, that the transfer phase of adaptation reinstantiates the same cortical manifold structure as observed during initial adaptation." Perhaps I am underestimating the audience but I think "cortical manifold structure" would be difficult to understand for most people, and indeed, some might interpret it in a completely different way to what is intended by the authors. 

To continue with the Results, Figure 1C, what do the surface map represent, and how are they constructed from centered covariance matrices? There is no explanation neither in text nor in the legend, and the color bar with "arbitrary units" does not help. It took me some time, after reading further and back, to realize that these maps probably represent the first PC of the corresponding epoch, although I could not match it to Figure 2 - are PC1s shown in Figure 2 representing the average of left and right hand PC1 from Figure 1?

Perhaps more easily interpretable (i.e. imbued with functional meaning) headings would help, e.g. "Figure 2: Manifold structure and eccentricity during Baseline trials." can be replaced with "Figure 2: Main functional connectivity networks during Baseline trials"?

Along the same lines, Figure 3 is very dense and the associated text is hard to parse.

The functional meaning of manifold eccentricity is hard to grasp, especially when considering one condition (e.g. baseline trials as in Figure 2). For instance, what does high eccentricity ("greater functional segregation") of visual cortex represent, in terms of underlying neuronal / structural connectivity? Does it imply that there is little connectivity between the visual cortex "network" and the rest of the brain? Similarly, what does the contralateral tuning of eccentricity mean, especially for the baseline periods (Figure 4)? Stronger eccentricity - stronger segregation? In other words, sensorimotor regions in the left hemisphere are less connected to the rest of the brain when using right hand (please see below)? How are the obvious consequences of the region-specific BOLD activity increase associated with contralateral hand movements factored in? Or are these activity changes subsumed under the manifold eccentricity? 

What does the preferential tuning of the eccentricity for the left hand in both hemispheres in precuneus(?) might indicate (posterior blue region in Figure 4E)?

Again, perhaps I am confused about the proper interpretation of manifold eccentricity changes, but the analysis in Figure 7 seems to contradict the interpretation of increased segregation: "Together, these results suggest that the manifold expansions of left and right M1 when using the contralateral hand mainly arise from their increased connectivity with other sensory-motor areas in cortex." It might be that the increased segregation "from other brain networks" can still be associated with increased connectivity a given "network", although the relative contribution of "within" vs "between" changes of connectivity still eludes me. 

The analysis in Figure 9 first states that "Specifically, for the DMN-A networks (which mainly encompass mPFC, PCC and angular gyrus), greater expansion along the manifold during early learning corresponded to lower error (i.e., improved performance).", but then concludes that "Collectively, these findings suggest that greater integration of the DMN-A network with several other higher-order cognitive networks generally corresponded to a more effective reduction of subjects' visuomotor errors." I admit that I am not following those seemingly contradictory statements, since the expansion is presumably increase in eccentricity, which has been interpreted as increase in segregation through the manuscript?

All in all, I think the meaning if eccentricity, supposed to serve as a "multivariate index" of connectivity, needs to be unpacked and clarified better, and perhaps the limitations of using a single number measure encompassing and averaging over enormous diversity of potential underlying effects should be discussed.

4) The valid point is made that only very few prior studies investigated similar paradigms with fMRI and they focused on more standard activation amplitude analysis within individual brain regions (but see Tzvi et al 2020, link below), while the present study goes beyond this work by looking at global whole-brain connectivity. But why there was no attempt to bridge the two types of analyses directly, in the same subjects, by analyzing also task-related activity using commonly used GLM approaches? I think this comparative approach would greatly enhance the scope and the interpretability of this work, and will help addressing the point (3) above.

Minor issues/clarifications:

"The target appeared at one of eight randomized locations, separated by 45° increments (0, 45, 90, 135, 180, 225, 270, and 315°), in bins of eight trials." 

- was there a learning dynamics within each "bin" of 8 trials with the same target (and each new target increased the error)? Or it was mainly gradual learning regardless of the target bin? What was the reason for 8-trial bins?

I think references and a discussion of the results of two relevant studies by Kirby et al and especially Tzvi et al are missing: 

https://pubmed.ncbi.nlm.nih.gov/31542802/

https://www.sciencedirect.com/science/article/pii/S1053811920306285

Page 4: "By disentangling effector-specific (right versus left hand) versus effector-independent changes in manifold structure, we found that it was predominantly the connectivity of DMN areas, and not sensorimotor regions, that were modulated across both the learning and transfer phases of the task." - grammar? (was / were) 

Page 24: "While significant changes in the manifold embedding of individual brain regions indicate an alteration in those regions' patterns of whole-brain connectivity, it is not obvious the precise nature of these connectivity alterations." - grammar? (the precise nature of these connectivity alterations is not obvious)

Page 34: "…and that inter-individual differences in the manifold embedding of DMN areas was related to both learning and generalization performance." - grammar? (was / were)

Lastly, will the entire preprocessing and code be made available? Given the elaborated and highly abstracted pipelines, I think it would be very good for reproducibility.

---

## [Decision Letter · Decision Letter 2]

Dear Jason,

Thank you for your patience while we considered your revised manuscript "Human cortical-subcortical manifold structure during the transfer of motor learning" for publication as a Research Article at PLOS Biology. This revised version of your manuscript has been evaluated by the PLOS Biology editors, the Academic Editor and the original reviewers.

Based on the reviews, we are likely to accept this manuscript for publication, provided you satisfactorily address the remaining points raised by the reviewers and the following data and other policy-related requests:

* We would like to suggest a different title to improve its accessibility for our broad audience: 

Transfer of motor learning between hands involves the same activity patterns in the default mode network as those that support initial learning

* Please include the license number(s) of the ethical approval for study.

* Please specify whether the participants provided written or verbal consent.

* DATA POLICY:

Regardless of the method selected, please ensure that you provide the individual numerical values that underlie the summary data displayed in the following figure panels as they are essential for readers to assess your analysis and to reproduce it: 1B, 3E, 4E, 5E, 6BD, 8ABCD, 9B, S3, S4, S6B and S7BE. 

CODE POLICY

We expect to receive your revised manuscript within two weeks. 

*Published Peer Review History*

*Press*

Sincerely,

Christian

Christian Schnell, PhD

Senior Editor

cschnell@plos.org

PLOS Biology

Reviewer #1 (Hiroshi Imamizu): The authors fully addressed my previous concerns. 

Reviewer #2: The authors have done a very nice job responding to my comments. I especially appreciated the new analyses, which offer additional support to some of the interpretations offered in the previous submission.

Reviewer #3: The authors meticulously and extensively addressed my concerns and considerably improved the manuscript. Congratulations on the great study!

Two minor remarks:

In the Introduction, I suggest to add the reference to a recent study on thalamic contributions to the motor learning, https://pubmed.ncbi.nlm.nih.gov/39542067/, to the sentence:

"However, motor learning and generalization engage widely distributed circuits spanning cortical, subcortical, and cerebellar areas [1,7-9]."

Some equations in Methods | Neuroimaging data analysis are corrupted.

---

## [Editor Report · Decision Letter 3]

Dear Jason,

Thank you for the submission of your revised Research Article "Transfer of motor learning is associated with patterns of activity in the Default Mode Network" for publication in PLOS Biology. On behalf of my colleagues and the Academic Editor, Alexander Gail, I am pleased to say that we can in principle accept your manuscript for publication, provided you address any remaining formatting and reporting issues. These will be detailed in an email you should receive within 2-3 business days from our colleagues in the journal operations team; no action is required from you until then. Please note that we will not be able to formally accept your manuscript and schedule it for publication until you have completed any requested changes.

PRESS

Sincerely, 

Christian

Christian Schnell, PhD

Senior Editor

PLOS Biology

cschnell@plos.org